# `Unlocker`: Disentangle the Deadlock of Learning from Label-noisy and Long-tailed Data

**Shu Chen**[1] , **Hongjun Xu**[1] , **Ruichi Zhang**[1] , **Mengke Li**[2] , **Yonggang Zhang**[3]
**Yang Lu**[1] [*], **Bo Han**[4] , **Yiu-ming Cheung**[4] , **Hanzi Wang**[1]

[1]Key Laboratory of Multimedia Trusted Perception and Efficient Computing
Ministry of Education of China, Xiamen University
[2]College of Computer Science and Software Engineering, Shenzhen University
[3]Hong Kong University of Science and Technology
[4]Hong Kong Baptist University
`{chenshu, xuhongjun}@stu.xmu.edu.cn`, `luyang@xmu.edu.cn`, `zhangyg@ust.hk`

## Abstract

In real world, the observed label distribution of a dataset often mismatches its true distribution due to noisy labels. In this situation, noisy labels learning (NLL) methods directly integrated with long-tailed learning (LTL) methods tend to fail due to a dilemma: NLL methods normally rely on unbiased model predictions to recover true distribution by selecting and correcting noisy labels; while LTL methods like logit adjustment depends on true distributions to adjust biased predictions, leading to a deadlock of mutual dependency defined in this paper. To address this, we propose `Unlocker`, a bilevel optimization framework that integrates NLL methods and LTL methods to iteratively disentangle this deadlock. The inner optimization leverages NLL to train the model, incorporating LTL methods to fairly select and correct noisy labels. The outer optimization adaptively determines an adjustment strength, mitigating model bias from over- or under-adjustment. We also theoretically prove that this bilevel optimization problem is convergent by transferring the outer optimization target to an equivalent problem with a closed-form solution. Extensive experiments on synthetic and real-world datasets demonstrate the effectiveness of our method in alleviating model bias and handling long-tailed noisy label data. Code is available at `https://github.com/ChenShu248/Unlocker`.

## 1 Introduction

Long-tailed noisy label learning (LTNLL) focuses on addressing the coexistence of long-tailed distribution and noisy labels in datasets. Existing LTNLL methods [1, 2, 3, 4, 5, 6] are commonly under the potential assumption that the observed long-tailed distribution based on the noisy labels is consistent with the true distribution of the clean labels. However, empirical observations from real world reveal that noisy labels can alter the original true distribution, leading to a deviation between the observed distribution and the true underlying distribution of the dataset [7, 8, 9]. Due to the diverse real-world noise patterns, the deviations of the distribution are various, typically including three situations: *consistent*, *relieve*, and *aggravate*, as illustrated in Figure 1a.

When addressing this issue of distribution deviation, neither noisy label learning (NLL) nor long tail learning (LTL) can be effective due to a *deadlock* dilemma of mutaul dependency. A mainstream in NLL highly rely on the model prediction to select and correct noisy labels to recover clean labels [10, 11, 12, 13, 14]. Yet in long-tailed scenarios, model predictions get biased towards head classes,

---

[*]Corresponding Author: Yang Lu

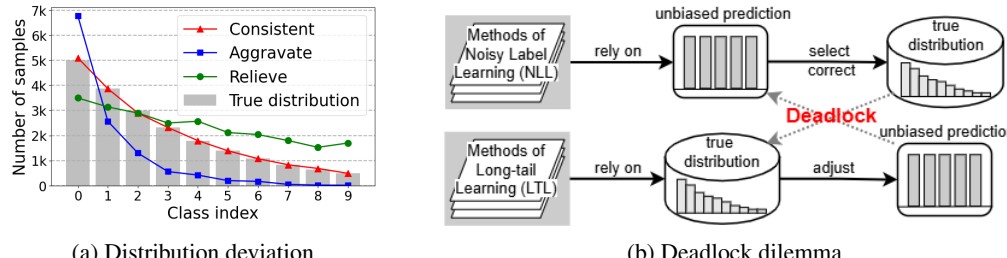

(a) Distribution deviation            (b) Deadlock dilemma

Figure 1: (a) Three typical scenarios of distribution deviation between the observed distribution and the true distribution. *Consistent*, *relieve*, and *aggravate* denote that the imbalance ratio (IR) of the observed distribution is identical to, lower than, and higher than the IR of the true distribution, respectively. (b) Deadlock between the noisy label learning (NLL) methods and the long-tailed learning (LTL) methods: NLL relies on the unbiased prediction to recover true distribution, while LTL requires the true distribution to recalibrate the biased prediction, creating a circular dependency.

posing challenges for selecting and correcting noisy samples in tail classes [1, 8]. Logit adjustment methods in LTL can recalibrate the biased predictions while relying on the true label distribution [15, 16, 17, 18, 19]. Due to the true distribution deviation, these methods fail to perform accurate adjustments. At this point, we can see a deadlock between NLL and LTL, as shown in Figure 1b. NLL relies on the unbiased predictions to restore a true distribution by selecting and correcting. Conversely, LTL can provide unbiased predictions but relies on a true distribution.

To disentangle the deadlock, we propose a novel method `Unlocker` based on the bilevel optimization framework. The core idea is jointly estimating the true distribution of the training set and optimizing an appropriate parameter $\tau$ for the adjusting strength. We define the inner problem as training a model using NLL adjusted by LTL, and the outer problem as optimizing the $\tau$ based on the inner problem's solution. In this framework, model predictions are adjusted using both $\tau$ and the estimated true distribution. This enables the model to better distinguish noisy labels. Improved noisy label selection and correction enhance distribution estimation accuracy. Accurate distribution estimation facilitates $\tau$ optimization, promoting model to reduce bias on the test set. This iterative process continues until convergence, gradually breaking the deadlock. Our main contributions are as follows:

- We define a new and challenging long-tailed noisy label problem of the deviation between the observed distribution based on noisy labels and the true distribution of the clean labels.

- We define a deadlock dilemma between the NLL and LTL, where their mutaul dependency renders both methods ineffecvtive.

- We propose a novel method `Unlocker` which bases on the bilevel optimization to effectively combine NLL and LTL by adaptively optimizing the adjustment strength.

## 2 Preliminaries

### 2.1 Noisy Label Selection

Noisy label learning has developed various methods to improve model performance by addressing label noise. A mainstream is to select noisy label samples from clean ones and correct the noisy labels [11, 12, 13, 14], which has been proven effective in noisy label learning tasks. Specifically, the selection of noisy labels relies on logit-based metrics, which are designed to maximize the discrimination between noisy and clean labels, enabling an efficient separation of the two subsets. Based on these metrics, the original dataset $\mathcal{D}$ is partitioned into a clean subset $\mathcal{D}_{clean}$ and a noisy subset $\mathcal{D}_{noisy}$. Then, the noisy labels in the $\mathcal{D}_{noisy}$ are corrected by leveraging techniques such as semi-supervised learning [20, 21]. The overall training objective is formulated as the total loss $L_{NLL}$, which is the sum of the clean loss $L_{clean}$ computed on $\mathcal{D}_{clean}$ and the noisy loss $L_{noisy}$ calculated on $D_{noisy}$. Conventionally, the cross-entropy loss is utilized for $L_{clean}$ to ensure accurate classification on reliable data. For $L_{noisy}$, more robust loss functions are often applied to reduce the impact of noise during training [22].

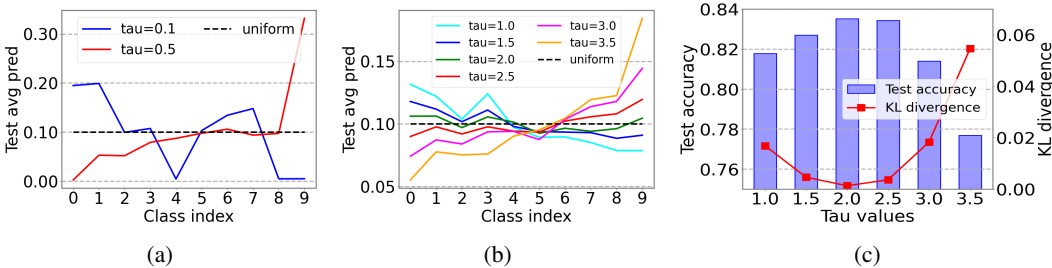

(a)                                         (b)                                         (c)

Figure 2: (a) Average prediction on CIFAR-10 test set of a NLL method DivideMix [11] adjusted by the true distribution prior $\mathbb{P}_{\text{train}}(y)$. The training set is CIFAR-10 with imbalance ratio (IR) of 50 and 40% symmetric noise. The model is highly sensitive to the $\tau$, making it prone to under-adjustment or over-adjustment. (b) Test average predictions of models trained using logit adjustment (LA) across different $\tau$ on a clean long-tailed CIFAR-10 dataset with IR of 50. As $\tau$ increases, the average predictions transition from skewing toward head classes to uniformity and then to skew toward tail classes. (c) Corresponding test accuracies and KL divergences (between test average predictions and a uniform distribution) for varying $\tau$. When $\tau = 2.0$ or $\tau = 2.5$, the trained model attains the highest accuracy and ouputs the most uniform average predictions.

## 2.2 Logit Adjustment

Logit adjustment [17, 23, 19] is an effective technique for long-tailed learning. In a standard classification with $C$ classes, the class posterior follows Bayes' theorem $\mathbb{P}(y|x) = \frac{\mathbb{P}(x|y)\mathbb{P}(y)}{\mathbb{P}(x)}$. Since $\mathbb{P}(x)$ is constant across all classes, it can be omitted. Thus, cross-entropy training yields $\mathbb{P}_{train}(y|x) \propto \mathbb{P}(x|y)\mathbb{P}_{train}(y)$, where $\mathbb{P}_{train}(y)$ is the class prior in training datasets. For balanced test datasets, a balanced prior $\mathbb{P}_{\text{bal}}(y) = \frac{1}{C}$ is desired, leading to $\mathbb{P}_{\text{bal}}(y|x) \propto \mathbb{P}(x|y)$. Assuming the likelihood $\mathbb{P}(x|y)$ is unchanged between the train and balanced datasets, we have the following relationship:

$$\mathbb{P}_{\text{test}}(y|x) \propto \frac{\mathbb{P}_{\text{train}}(y|x)}{\mathbb{P}_{\text{train}}(y)} \propto \text{softmax}(\boldsymbol{\theta}_y(x) - \log \mathbb{P}_{\text{train}}(y)), \tag{1}$$

where $\boldsymbol{\theta}_y(x)$ is the model's logit for class $y$. To further improve flexibility, a temperature parameter $\tau$ which modulates the adjustment intensity is introduced, leading to the general LA formulation:

$$\arg\max_{y \in [C]} \text{softmax}(\boldsymbol{\theta}_y(x) - \tau \cdot \log \mathbb{P}_{\text{train}}(y)). \tag{2}$$

Typically, $\tau > 0$ is used for post-hoc adjustment, while $\tau < 0$ can be incorporated as logit adjustment loss during training.

## 3 `Unlocker`: Breaking the Deadlock with Bilevel Optimization

### 3.1 Motivation

To break the deadlock, we first attempt to adjust the logits of a classical NLL method DivideMix [11] using the true distribution prior $\mathbb{P}_{\text{train}}(y)$. Ideally, incorporating $\mathbb{P}_{\text{train}}(y)$-based adjustments leads to unbiased confidence across all classes, reflected by a uniform average prediction on the test set, as depicted by the black line in Figure 2a. However, directly employing $\mathbb{P}_{\text{train}}(y)$ for LA results in suboptimal model performance due to under-adjustment or over-adjustment [24]. Specifically, as $\mathbb{P}_{\text{train}}(y)$ is the same, when the parameter $\tau = 0.5$ (red line), the tail classes are over-adjusted, causing the average prediction to skew towards the tail and damaging the head, as shown in Figure 2a. Conversely, when $\tau = 0.1$ (blue line), the adjustments are insufficient, remaining the predictions biased towards the heads. Without an appropriate $\tau$ to modulate the adjusting strength, even with the precise $\mathbb{P}_{\text{train}}(y)$, the model could not achieve an ideal unbiased state.

To further investigate $\tau$ in modulating model bias, we eliminate the influence of noisy label and implement a simple control variable experiment on a clean long-tailed dataset. We test different $\tau$ of $\{1.0, 1.5, 2.0, 2.5, 3.0, 3.5\}$ with the same $\mathbb{P}_{train}(y)$. As shown in Figure 2b, when $\tau = 1.0$, the tail classes are under-adjusted. As $\tau$ increasing, the test average predictions become more uniform.

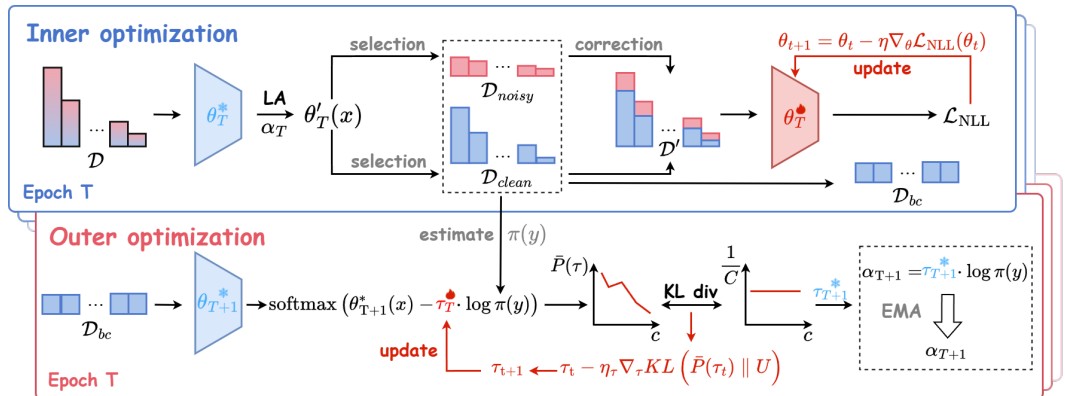

Figure 3: The illustration of the proposed method `Unlocker` based on the bilevel optimization framework, which iteratively disentangle the deadlock between the noisy label learning (NLL) and long-tailed learning (LTL). In epoch $T$, the inner optimization trains the model $\theta$ using NLL integrated with logit adjustment in LTL. By adjusting logits as $\theta'_T(x)$, it fairly selects and corrects noisy labels, and trains $\theta$. The outer optimization adaptively updates the adjusting strength parameter $\tau$ to modulate model bias towards balance. An EMA strategy is introduced to stabilize the adjustments after the whole optimizations completing. Parameters frozen during optimization are denoted by the snowflake (no gradient flow), while trainable parameters are marked by fire.

When $\tau = 2.0$ or $\tau = 2.5$, the average predictions are closest to uniformity, and the model achieved the highest test accuracy as depicted in Figure 2c. When further increasing $\tau$, the model gets over-adjusted, and the predictions skew towards the tail. This simple experiment demonstrates that an appropriate $\tau$ is crucial for achieving unbiased model state and influences the test accuracy.

## 3.2 Bilevel Optimization

Based on the motivation, we propose `Unlocker` which utilizes the bilevel optimization framework to iteratively disentangle the deadlock. The inner layer optimization trains model using NLL integrating LA, and the outer layer optimizes a learnable parameter $\tau$ to regulate the strength of adjustments. Through gradient coupling between layers, this framework maintains model capability against noisy labels while dynamically calibrating class bias.

### 3.2.1 Inner Optimization

Given a long-tailed noisy label training set $\mathcal{D} = \{(x_i, y_i)\}_{i=1}^N$, the inner layer employs NLL methods integrating LA to train model $\theta$. At epoch $T$, for the noisy label selection, an adjustment $\alpha_T$ (post-hoc) is applied to the row logits $\theta_T(x)$ (gradient-free) to achieve fairer noise recognition, particularly enhancing the recognition accuracy of the tail classes:

$$\theta'_T(x) = \theta_T(x) - \alpha_T. \tag{3}$$

Based on the adjusted logits $\theta'_T(x)$, selecting metrics of NLL methods are computed to divide $\mathcal{D}$ into $\mathcal{D}_{clean}$ and $\mathcal{D}_{noisy}$. For the noisy label correction for $\mathcal{D}_{noisy}$, we follow the method in [25] to apply post-hoc adjustment on the generation of corrected labels. With $\mathcal{D}_{clean}$ and corrected $\mathcal{D}_{noisy}$, the model training loss $\mathcal{L}_{NLL}$ can be computed and the model $\theta_T$ updated at step $t$ is given by:

$$\theta_{t+1} = \theta_t - \eta \nabla_\theta \mathcal{L}_{NLL}(\theta_t), \tag{4}$$

where $\eta$ is the learning rate. Upon completing $T$ epochs, the optimal inner parameters $\theta^*_{T+1}$ are fixed as the inner solution for the outer optimization.

Before diving into the outer optimization, two critical components are required in addition to support the outer object: (i) the construction of a balanced and clean subset $\mathcal{D}_{bc}$, and (ii) the estimation of the training set class prior $\pi(y)$.

**Construct $\mathcal{D}_{bc}$.** During the process of noisy label selection, we further filter the top $q\%$ of samples with the highest confidence of clean as a high-purity clean subset. As the clean subset follows a long-tailed distribution, we apply over-sampling and form the final balanced clean subset $\mathcal{D}_{bc}$.

**Estimate $\pi(y)$.** As $\mathbb{P}_{train}(y)$ is unknown in our long-tailed noisy labels problem, we estimate a proxy distribution prior $\pi(y)$ to approximate it. After the correction for $\mathcal{D}_{noisy}$ completing, we estimate $\pi(y)$ based on the clean labels of the $\mathcal{D}_{clean}$ and the corrected labels of the $\mathcal{D}_{noisy}$.

### 3.2.2 Outer Optimization

The outer optimizes a learnable parameter $\tau$ to dynamically regulate the adjusting strength to modulate model bias towards balance. The core objective is to minimize the discrepancy between the model's class confidence and a uniform distribution. To measure model bias, we leverage the balanced clean validation subset $\mathcal{D}_{bc}$ constructed in 3.2.1. Given the fixed inner parameters $\theta_{T+1}^*$, the model's logits on $\mathcal{D}_{bc}$ are $\theta_{T+1}^*(x)$ (gradient-free), and the $\tau_T$-regulated prediction distribution is:

$$P(y \mid x; \tau_T) = softmax\left(\theta_{T+1}^*(x) - \tau_T \cdot \log(\pi(y))\right), \tag{5}$$

where $\log(\pi)$ is the fixed class prior term estimated in Sec. 3.2.1. Averaging $P(y \mid x; \tau_T)$ over $\mathcal{D}_{bc}$ yields the class-averaged prediction $\bar{P}(\tau_T) = \mathbb{E}_{x \sim \mathcal{D}_{bc}} P(y \mid x; \tau_T)$, leading to the outer objective:

$$\min_\tau KL\left(\bar{P}(\tau_T) \parallel U\right), \quad U_c = \frac{1}{C} \ (\forall c \in \{1, \dots, C\}), \tag{6}$$

where $U$ is a uniform prediction tensor, $C$ is the number of classes. Minimizing this divergence adapts $\tau_T$ to drive $\bar{P}(\tau_T)$ toward uniformity, forcing balanced class confidence on the test set. The update of $\tau_T$ at step $t$ is derived via gradient descent:

$$\tau_{t+1} = \tau_t - \eta_\tau \nabla_\tau KL\left(\bar{P}(\tau_t) \parallel U\right). \tag{7}$$

After completing the optimization of $\tau_T$, the adjustments are calculated as $\alpha_{T+1} = \tau_{T+1} \cdot \log \pi(y)$. To ensure the stability of the adjustments during the training, especially when the $\pi(y)$ fluctuates in the early training, we adopt an Exponential Moving Average (EMA) strategy to update the adjustments. At the $T$-th epoch, we update the adjustments as follows:

$$\alpha_{T+1} = \beta \cdot \alpha_{T+1} + (1 - \beta) \cdot \alpha_T, \tag{8}$$

where $\beta$ is a hyperparameter that controls the decay rate of the EMA. $\alpha_{T+1}$ is then passes into epoch $T+1$ to adjust the inner-model training.

### 3.2.3 Training Overview

As illustrated in Figure 3, through the bilevel optimization, we effectively combine the NLL method and LA to handle long-tailed noisy label training data, breaking the mutual dependency. First, in inner optimization, we perform post-hoc logit adjustment on the selection of the NLL methods, enabling a more fair selection of noisy labels in tail classes. Based on the selection results, we construct a balanced clean subset $\mathcal{D}_{bc}$ and partition the training set into $\mathcal{D}_{clean}$ and $\mathcal{D}_{noisy}$. Next, we use the NLL methods to correct the noisy labels in $\mathcal{D}_{noisy}$, and estimate the $\pi(y)$. Then, the model is trained and adjusted using the corrected labels. In outer optimization, the $\tau$ is optimized to minimize the discrepancy between model preference and uniform distribution based on the $\mathcal{D}_{bc}$. After the optimization of $\tau$ is completed, we update the adjustments through EMA to ensure the stability of the training. The detailed pseudocode of the training process can be found in the Appendix A.1.

### 3.3 Theoretical Analysis

In this section, we establish the convergence of our bilevel optimization framework by deriving a closed-form approximation for the outer optimization of $\tau$. Specifically, we first prove that the outer objective is differentiable and exists a global minimum. Then, leveraging the uniform target distribution, we show that the optimal $\tau$ can be approximated via least-squares minimization, reducing the bi-level problem to a normal optimization. This reduction guarantees the convergence of our method. The detailed processes of proof are provided in the Appendix A.2.

**Theorem 1** (Differentiability and Gradient Descent Applicability for Outer Optimization). *Under the assumptions that $\theta^*(x)$ is continuously differentiable with respect to $\tau$ and $\pi_c(y) > 0$ for all classes $c \in \{1, 2, \dots, C\}$, the outer objective function $J(\tau) = KL\left(\bar{P}(\tau) \parallel U\right)$ is continuously differentiable with respect to $\tau$. The gradient descent update rule $tau_{t+1} = \tau_t - \eta_\tau \nabla_\tau J(\tau_t)$ ensures that $J(\tau)$ decreases monotonically along the negative gradient direction.*

Based on the Theorem 1, we can further prove the existence of a global minimum.

**Lemma 1** (KL Divergence Bounds). *By the definition of KL divergence, the objective function $J(\tau) = KL(\bar{P}(\tau)\|U)$ establishes the following lower bound and upper bound:*

$$0 \leq J(\tau) = \log C - H(\bar{P}(\tau)) \leq \log C.$$

**Theorem 2** (Existence of Global Minimum). *Given the continuity of $J(\tau)$ and its lower bound $J(\tau) \geq 0$ and upper bound $\lim_{\tau \to \pm\infty} J(\tau) = \log C$ in Lemma 1, and $\lim_{\tau \to \pm\infty} J(\tau) = \log C$, there exists at least one global minimum point $\tau^* \in \mathbb{R}$.*

In summary, the objective function $J(\tau)$ is bounded and has a global minimum, providing a theoretical guarantee for the optimization process. Leveraging the uniform prior $U$, we derive a closed-form approximation for $\tau$ via least-squares minimization of the residual function.

**Proposition 1** (Least-Squares Closed-Form Solution). *Define $\mu_c = \frac{1}{N}\sum_x \theta_c(x)$, the least-squares closed-form solution for $\tau$ is:*

$$\tau^{LS} = \frac{\sum_c (\log \pi_c(y) - \log \pi_1)(\mu_c - \mu_1)}{\sum_c (\log \pi_c(y) - \log \pi_1)^2}. \tag{9}$$

This $\tau^{LS}$ reduces the bi-level optimization problem to a single-level optimization, establishing the theoretical convergence guarantee for our framework.

## 4 Experiments

### 4.1 Experimental Setup

**Datasets.** We conduct simulated experiments on CIFAR-10/100[26], including three cases: *consistent*, *relieve*, and *aggravate* which we summarize in Figure 1a . CIFAR-10/100 contains 50,000 training images and 10,000 test images of size $32 \times 32$ pixels, where CIFAR-10 contains 10 classes and CIFAR-100 contains 100 classes. To simulate all scenarios, we first construct a long-tailed dataset with an imbalance ratio (IR) by exponential decay. $IR = N_{max}/N_{min}$ represents the ratio of the number of samples of the majority class to the number of samples of the minority class.

Before generating label noise, due to the real-world noise patterns are complex and diverse, we systematically explore the effects of existing noise addition methods on the imbalance ratio of the long-tailed dataset. The detailed analysis results refer to the Appendix A.4. We find that the joint noise hardly changes the original long-tailed distribution in the case of low imbalance ratio, the symmetry noise alleviates the imbalance ratio, and the t2h noise aggravates. Based on these findings, we introduce the joint noise to the clean long-tailed dataset to simulate the *consistent* case, use the symmetry noise to simulate the *relieve* case, and inject the t2h noise to simulate the *aggravate* case. The detailed noise addition methods refer to the Appendix A.3. In the following experiments, the imbalance rate is selected in $\{10, 50, 100\}$, and the noise rate $\eta$ which describes the proportion of label noise in the training dataset is set to $\{0.4, 0.6\}$.

We also evaluate the performance of our method on real-world datasets, including Red Mini-ImageNet [27], Clothing1M [28] and WebVision-50 [29]. Red Mini-ImageNet contains 100 classes with 50,000 training images and 5,000 testing images, annotated via controlled noise protocols. Clothing1M contains 1 million training images obtained from online shopping websites, with 50k, 14k, and 10k images split into clean labels for training, validation, and testing across 14 classes. WebVision consists of 2.4 million images from Google and Flickr, sharing the same 1,000 categories as ImageNet, and includes 50,000 human-annotated validation and test images. Following the experimental setting of [11], we use the first 50-class subset (WebVision-50) for training and evaluate on both WebVision validation set and ILSVRC12 validation set for the same 50 classes.

**Baselines.** We compare our method with the following four categories of methods: (1) Noisy label learning (NLL) methods: including DivideMix [11], UNICON [12] and DPC [14]. (2) NLL methods with post-hoc LA based on real distribution (NLL+LA post-hoc). (3) NLL methods with LA based on real distribution during training (NLL+LA). (4) Long-tailed noisy label learning methods (LTNLL), including RoLT [1], TABASCO [8] and DaSC [4]. We select the baseline NLL methods to combine with our method. Additionally, for real-world dataset, we further include baselines: MentorNet [30], Co-teaching [10], HAR [31], UCL [2], RCAL [3], GSS [5].

Table 1: Test accuracy (%) comparison of different methods on the CIFAR-100 dataset under varying imbalance ratios (IR), noise types and noise rates $\eta$, involving three scenarios of true distribution shifts. * denotes results from the original papers. Green numbers indicate improvements of our method combined with the NLL method over the original NLL method. Boldface represents the best performance in each case.

| Dataset | CIFAR-100 | | | | | | | | | |
|---|---|---|---|---|---|---|---|---|---|---|
| Types | consistent | | relieve | | | | aggravate | | | |
| IR | 10 | | 50 | | 100 | | 10 | | 50 | |
| $\eta$ | joint.40% | joint.60% | sym.40% | sym.60% | sym.40% | sym.60% | t2h.40% | t2h.60% | t2h.40% | t2h.60% |
| RoLT [1] | 32.57 | 21.44 | 22.68 | 14.76 | 23.51* | 16.61* | 32.65 | 22.12 | 20.28 | 14.08 |
| TABASCO [8] | 55.31 | 45.40 | 40.91 | 31.28 | 36.91* | 26.25* | 45.37 | 43.60 | 32.18 | 17.53 |
| DaSC [4] | 58.06 | 45.64 | 41.96 | 32.17 | 36.40 | 29.59 | 55.67 | 39.93 | 34.03 | 13.47 |
| DivideMix [11] | 68.96 | 62.54 | 50.75 | 39.56 | 45.13 | 37.15 | 63.24 | 65.76 | 51.53 | 48.68 |
| DivideMix+LA (post-hoc) | 68.53 | 62.74 | 32.37 | 17.53 | 23.63 | 11.89 | 70.18 | 68.95 | 52.41 | 52.68 |
| DivideMix+LA | 68.53 | 63.87 | 16.30 | 5.31 | 7.31 | 2.51 | 68.67 | 63.94 | 46.56 | 42.93 |
| DivideMix+Unlocker | 71.95 | 68.97 | 60.38 | 53.34 | 54.26 | 45.07 | 72.20 | 69.05 | 62.59 | 58.07 |
| vs. DivideMix | ↑2.99 | ↑6.43 | ↑9.63 | ↑16.98 | ↑9.13 | ↑7.92 | ↑8.96 | ↑3.29 | ↑11.06 | ↑9.39 |
| UNICON [12] | 63.63 | 62.42 | 52.16 | 44.53 | 46.82 | 39.83 | 60.37 | 58.61 | 49.60 | 40.53 |
| UNICON+LA (post-hoc) | 67.15 | 63.13 | 53.18 | 45.27 | 46.84 | 40.44 | 65.41 | 58.81 | 50.32 | 42.78 |
| UNICON+LA | 62.39 | 61.40 | 8.53 | 50.19 | 2.75 | 1.80 | 67.58 | 58.64 | 55.94 | 41.67 |
| UNICON+Unlocker | 69.11 | 65.53 | 54.83 | 52.98 | 48.38 | 44.87 | 65.97 | 60.49 | 56.18 | 44.70 |
| vs. UNICON | ↑5.48 | ↑3.11 | ↑2.67 | ↑8.42 | ↑1.56 | ↑5.04 | ↑5.60 | ↑1.88 | ↑6.58 | ↑4.17 |
| DPC [14] | 70.81 | 54.91 | 44.66 | 34.84 | 39.43 | 23.71 | 70.03 | 11.32 | 36.73 | 1.04 |
| DPC+LA (post-hoc) | 69.97 | 55.13 | 39.77 | 30.88 | 39.59 | 21.62 | 70.05 | 22.85 | 40.67 | 2.44 |
| DPC+LA | 70.01 | 50.24 | 18.16 | 5.37 | 5.33 | 2.38 | 69.88 | 61.76 | 28.36 | 3.21 |
| DPC+Unlocker | 71.86 | 66.78 | 57.19 | 50.01 | 50.18 | 45.31 | 72.09 | 65.78 | 55.34 | 49.92 |
| vs. DPC | ↑1.05 | ↑11.84 | ↑12.53 | ↑15.17 | ↑10.75 | ↑21.60 | ↑2.06 | ↑54.46 | ↑18.61 | ↑48.88 |

Table 2: Top 1 and Top 5 test accuracy on Webvision and ImageNet validation sets. The best results are bolded. The experimental results of other methods are from [5].

| Test | WebVision | | ILSVRC12 | |
|---|---|---|---|---|
| | Top-1 | Top-5 | Top-1 | Top-5 |
| Standard | 62.5 | 80.8 | 58.50 | 81.8 |
| Co-teaching [10] | 63.58 | 85.20 | 61.48 | 84.70 |
| MentorNet [30] | 63.00 | 81.40 | 57.80 | 79.92 |
| HAR [31] | 75.5 | 90.7 | 70.3 | 90.0 |
| RoLT+ [1] | 77.64 | 92.44 | 74.64 | 92.48 |
| RCAL [3] | 76.24 | 92.83 | 73.60 | 93.16 |
| GSS [5] | 83.64 | 95.86 | **74.17** | 95.22 |
| DivideMix+Unlocker | **83.79** | **96.55** | 73.21 | **96.37** |

Table 3: Test accuracy (%) on Red-Mini-Imagenet dataset. The best results are bolded. The experimental results of other methods are from [5].

| IR | 10 | | 100 | |
|---|---|---|---|---|
| $\eta$ | 20% | 40% | 20% | 40% |
| ERM | 40.42 | 31.46 | 30.88 | 31.46 |
| DivideMix [11] | 48.76 | 48.96 | 33.00 | 34.72 |
| UNICON [12] | 40.18 | 41.64 | 31.86 | 31.12 |
| HAR [31] | 46.61 | 38.71 | 32.60 | 31.30 |
| ULC [2] | 48.12 | 47.06 | 34.24 | 34.84 |
| TABASCO [8] | 50.20 | 49.68 | 37.20 | 37.12 |
| GSS [5] | 52.33 | 50.91 | 40.25 | 36.58 |
| DivideMix+Unlocker | **53.29** | **51.48** | **41.51** | **37.14** |

**Implementation Details.** To ensure a fair comparison with existing methods, we keep the training configurations consistent with the baseline NLL methods. Specifically, we employ the 18-layer ResNet as the backbone architecture. The mini-batch size is fixed at 256. All models are optimized using SGD with a momentum of 0.9. A random seed of 123 is used across all experiments to ensure reproducibility. For NLL methods combined with LA in baselines, the $\tau$ is set to 1.0. For our proposed method Unlocker, the learnable parameter $\tau$ is initialized to 1.0 and optimized using SGD with a momentum of 0.9. The initial learning rate for $\tau$ is set to 0.1, which is adjusted to 0.01 at the 150-th epoch. The $\beta$ in EMA to update adjustments is set to 0.9. All experiments are executed on a GeForce RTX 3090 GPU using the PyTorch 1.8.0 framework to maintain hardware consistency.

## 4.2   Results on Simulated Scenarios

We conduct a comprehensive performance comparison across three scenarios of the distribution deviations in Table 1. Our method combines with the NLL methods (NLL+Unlocker) demonstrates superiority over the LTNLL methods under all three conditions. Regarding NLL methods, integrating Unlocker yields substantial accuracy improvements. On CIFAR-100, the combination achieves performance boosts ranging from 1.05% to 54.46%. In contrast, NLL methods directly combined

Table 4: Test accuracy (%) comparison of different methods on the Clothing1M dataset The best results are in bold. The experimental results of other methods are from [5].

| Methods | CE | MentorNet [30] | Co-teaching [10] | DivideMix [11] | ULC [2] | RCAL+ [3] | GSS [5] | DivideMix+Unlocker |
|---------|-----|----------------|------------------|----------------|---------|-----------|---------|---------------------|
| Accuracy | 65.42 | 67.25 | 67.94 | 74.76 | 74.87 | 74.97 | 75.83 | **76.94** |

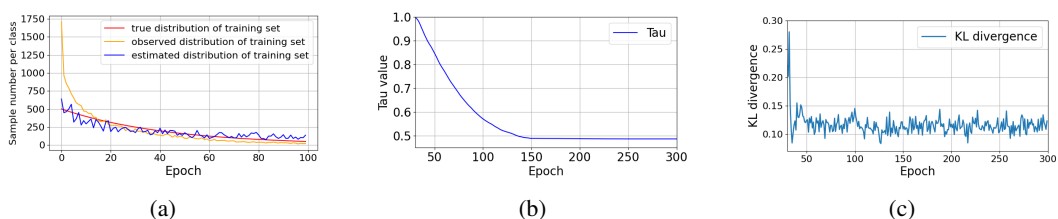

(a)                                                    (b)

Figure 4: Bias comparison of different models trained on CIFAR-100 (IR=100, sym.40%) (a) KL divergence between the average predictions on CIFAR-100 test set and a uniform prediction during training. (b) The average predictions in the last epoch on CIFAR-100 test set.

(a)                                    (b)                                    (c)

Figure 5: (a) The estimated training set distribution at the last epoch on CIFAR-100 with an imbalance ratio (IR) of 10 and t2h.60%. (b) The convergence trajectory of the parameter $\tau$ during training. (c) The KL divergence between the average predictions on the balanced clean subset $\mathcal{D}_{bc}$ and the CIFAR-100 test set. (b) and (c) are conducted on CIFAR-100 with IR=100 and sym.40%.

with true-distribution-based LA (both post-hoc and training-time LA) often suffer from the problem of over-adjustment, causing training collapse where predictions concentrate on a single class. Our method adaptively adjusts the parameter $\tau$ during training, effectively alleviating the issues of over- or under-adjustment. This mechanism enables a collaboration with NLL methods, jointly addressing the challenges of long-tailed noisy labels data. As a result, our method achieves the best performance among all methods, showing its robust adaptability to varying imbalance ratios and noise types. The results of the comparison experiment on CIFAR-10 are presented in the Appendix A.5.

### 4.3 Results on Real-world Datasets

We performed comparative experiments on three real-world datasets to validate the robustness of our method in practical setting. As shown in Table 2, DivideMix+Unlocker achieves 83.79% Top-1 and 96.55% Top-5 accuracy on the WebVision-50 validation set, outperforming all baselines. On the ILSVRC12 validation set, it achieves the highest 96.37% Top-5 accuracy and competitive 73.21% Top-1 accuracy. For Red-Mini-Imagenet, as shown in Table 3, our method attains 53.29% accuracy under IR=10 and $\eta$=20%, outperforming SOTA (52.33%) and improving upon DivideMix by 4.5%. Under conditions (IR=100, $\eta$=40%), it maintains 36.37% accuracy. On Clothing1M, as shown in Table Table 4, our method outperforms other methods by achieving 76.94% accuracy. These experiments fully verified the ability of our method in real-world long-tailed noisy label scenarios, enabling NLL methods to achieve a more uniform state against long-tail bias.

### 4.4 Effectiveness Study and Discussions

**Average Prediction Analysis.** To validate the effectiveness of our method in reducing models bias to achieve a more uniform state, we conduct experiments by training the models of different methods on CIFAR-100 training set with $IF = 100$ and $\eta = sym.40\%$, and outputting models' average predictions on the CIFAR-100 test set. In Figure 4a, during training, the KL divergence between the

test average prediction of the DivideMix and the uniform prediction stabilizes at a relatively high value, as depicted by the orange line. This means the model's average prediction is still far away from uniform, indicating persistent bias of the model. In contrast, by combining with our method `Unlocker`, the KL divergence down to approximately 0.08, signifying that the model's test average predictions approach uniformity and the model gets more unbiased. Figure 4b demonstrates the average prediction in the last epoch of the different model. As the orange bars showing, the test average predictions of the DivideMix skew toward head classes, neglecting tail classes. Conversely, our method's prediction (blue bars) distributes evenly across classes. These results collectively validate that our method effectively mitigates long-tailed impact and promotes balancer model state.

**Effectiveness of Distribution Estimation.** To verify the effectiveness of the distribution estimation module in our method, we conduct experiments on CIFAR-100 with IR = 10 and t2h.60%. As shown in the Figure 5a, the red line denotes the true distribution of the training set, the orange line represents the observed distribution based on noisy labels, and the blue line is the training set distribution estimated by our method. Notably, the imbalance ratio of the observed distribution is significantly higher than that of the true distribution, indicating that noisy labels aggravate the imbalance ratio of the true distribution. Our method estimates the distribution (blue line) that closely aligns with the true distribution (red line), demonstrating its capability to accurately capture the training set distribution.

**Convergence of $\tau$.** We demonstrate the convergence of the parameter $\tau$ for our method trained on the CIFAR-100 dataset with IR = 100 and sym.40%. As shown in the Figure 5b, our method achieves the stable convergence of the parameter $\tau$. Specifically, the optimization of $\tau$ starts after the warm-up of 30 epochs. In the first 100 epochs, the value of $\tau$ decreases rapidly. After 100 epochs, the descent rate slows down, and it converges to a value near 0.48 around 150 epochs.

**Effectiveness of $\mathcal{D}_{bc}$.** The balanced clean subset $\mathcal{D}_{bc}$ serves as a validation set to measure the model bias. Thus it is critical that $\mathcal{D}_{bc}$ can reflect the model's class confidence. To validate this, we conduct experiments on CIFAR-100 with IR = 100 and sym.40%, tracking the average predictions of the model on $\mathcal{D}_{bc}$ and the test set at each epoch, By computing their KL divergence, as shown in the Figure 5c, the discrepancy remains at a relatively low level of approximately 0.12. While there are slight fluctuations, they are in an acceptable small range. This result indicates that the average predictions on $\mathcal{D}_{bc}$ closely align with those on the test set, demonstrating that $\mathcal{D}_{bc}$ reflects model bias.

## 5 Related Work

Long-tailed and noisy label learning faces challenges in distinguishing clean samples from noisy ones in tail classes. RoLT [1] proposes a prototype-based noise detection method using class centroid distances to select noisy samples. ULC [2] combines class-specific noise modeling with uncertainty quantification to account for cognitive and incidental uncertainties. TABASCO [8] employs a weighted JS divergence and adaptive centroid distance to distinguish. These methods often integrate semi-supervised learning to refine predictions after noise detection. HAR [31] applies heteroscedastic adaptive regularization to high-uncertainty and low-density data points. RCAL [3] leverages unsupervised contrastive learning to eliminate noisy samples and restore representation distributions, enhancing model generalization. Detailed related work on long-tailed learning and noisy label learning individually is provided in the Appendix A.6.

## 6 Conclusion

In this paper, we address the challenging problem of long-tailed noisy label learning where the observed distribution based on noisy label deviates from the true distribution. When addressing this issue, a deadlock dilemma between noisy label learning (NLL) and long tail learning (LTL) arises. To disentangle the deadlock and tackle the long-tailed noisy label problem, we propose `Unlocker`, a bilevel optimization framework that iteratively optimizes an adjustment strength parameter $\tau$ to effectively combine the NLL methods and LTL methods. Extensive experiments on synthetic and real-world datasets demonstrate that `Unlocker` significantly outperforms SOTA methods. However, our method has limitations: (i) potential impurity of $\mathcal{D}_{bc}$, and (ii) approximation errors in distribution estimation. Our future research will focus on address these limitations.

## Acknowledgements

This study was supported in part by the National Natural Science Foundation of China under Grants 62376233, 62306181 and 62376235; in part by the Natural Science Foundation of Fujian Province under Grant 2024J09001; in part by the RGC Young Collaborative Research Grant C2005-24Y; in part by the NSFC / Research Grants Council (RGC) Joint Research Scheme under Grant N_HKBU214/21; in part by the General Research Fund of RGC under the Grants 12201323 and 12200725; in part by the RGC Senior Research Fellow Scheme under the Grant SRFS2324-2S02; and in part by Xiaomi Young Talents Program. YGZ was funded by Inno HK Generative AI R&D Center.

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

# A    Appendix / Supplemental Material

## A.1    Detailed Training Process

The following is the process of our proposed method `Unlocker`, which leverages a bilevel optimization framework. Within each epoch, the inner optimization employs NLL methods with LA to train model, while the outer loop optimizes the learnable parameter $\tau$ to dynamically scale the strength of LA. This process adaptively tunes $\tau$ to integrate NLL methods and LA, iteratively disentangling the NLL-LTL deadlock and enhancing model robustness against long-tailed noisy label data.

---

**Algorithm 1** Detailed training process of `Unlocker`

---

**Input:** Training data $\mathcal{D} = \{(x_i, y_i)\}_{i=1}^N$, Number of epochs $N_{epoch}$, NLL algorithm $\mathcal{A}$, learning rates$\eta_\tau$, EMA decay $\beta$

**Output:** Trained model $\theta$, Optimized $\tau$

**Initialize:** Model Parameters $\theta_0$, Learnable Parameter $\tau_0 \leftarrow 1.0$, Class Prior $\pi(y) \leftarrow$ EstimatePrior$(\mathcal{D})$, adjustments $\alpha \leftarrow \tau_0 \cdot \log(\pi(y))$

**for** $epoch = T$ from 1 to $N_{epoch}$ **do**

    // Inner Optimization

    **for** $k = 1$ to $K$ **do**

        $\mathcal{D}_k \leftarrow$ GetBatch$(\mathcal{D}, k)$

        **Inference Mode:** $\theta'_T \leftarrow \theta_T(\mathcal{D}_k) - \alpha_T$                     ▷ Post-hoc logit adjustment

    **end for**

    $\mathcal{D}_{clean}, \mathcal{D}_{noisy}, \mathcal{D}_{bc} \leftarrow \mathcal{A}(\theta'_T)$               ▷ Select noisy labels according to $\mathcal{A}$

    $\mathcal{D}_{noisy} \leftarrow \mathcal{A}(\theta'_T, \mathcal{D}_{noisy})$                 ▷ Correct noisy labels according to $\mathcal{A}$

    $\pi(y) \leftarrow$ EstimatePrior$(\mathcal{D}_{clean}, \mathcal{D}_{noisy})$           ▷ Estimate class prior of $\mathcal{D}$

    **for** $k = 1$ to $K$ **do**

        $\mathcal{D}_k \leftarrow$ GetBatch$(\mathcal{D}, k)$

        $\theta'_T \leftarrow \begin{cases} \theta_T + \alpha_T, & \text{if } \mathcal{D}_k \subset \mathcal{D}_{\text{clean}} \\ \theta_T, & \text{otherwise} \end{cases}$             ▷ logit adjustment

        $\mathcal{L}_{NLL} \leftarrow \mathcal{A}(\theta'_T(x), \mathcal{D}_k)$               ▷ Compute model loss according to $\mathcal{A}$

        $\theta_{k+1} \leftarrow \theta_k - \eta \nabla_\theta \mathcal{L}_{NLL}(\theta_k)$                     ▷ Update model

    **end for**

    // Outer Optimization

    **for** $k = 1$ to $B$ **do**

        $\mathcal{D}_k \leftarrow$ GetBatch$(\mathcal{D}_{bc}, k)$

        **Inference Mode:** $\theta_{T+1}(\mathcal{D}_k)$

        $\theta'_{T+1}(\mathcal{D}_k) \leftarrow \theta_{T+1}(\mathcal{D}_k) - \tau_k \cdot \log \pi(y)$           ▷ Post-hoc logit adjustment

        $\mathcal{L}_\tau \leftarrow KL\left(\mathbb{E}_{x \sim D_k} \text{softmax}(\theta'_{T+1}(\mathcal{D}_k)) \parallel U\right)$   ▷ Compute outer loss according to Eq. 6

        $\tau_{k+1} = \tau_k - \eta_\tau \nabla_\tau \mathcal{L}_\tau(\tau_k)$                       ▷ Optimize $\tau$

    **end for**

    $\alpha_{T+1} \leftarrow \tau_{T+1} \cdot \log(\pi(y))$                       ▷ Compute adjustments

    $\alpha_{T+1} \leftarrow \beta \cdot \alpha_{T+1} + (1 - \beta) \cdot \alpha_T$             ▷ EMA update for adjustments

**end for**

**return** $\theta, \tau$

---

## A.2    Theoretical Proof

*Proof of Theorem 1.* We prove the differentiability of the outer optimization with the following steps:

**Step 1: Objective Function Expansion** Substituting $\bar{P}(\tau_T) = \mathbb{E}_{x \sim \mathcal{D}_{bc}} P(y \mid x; \tau_T)$ into the KL

divergence $J(\tau) = KL\left(\bar{P}(\tau) \parallel U\right), U_c = \frac{1}{C}$, we get:

$$
\begin{aligned}
J(\tau) &= KL\left(\bar{P}(\tau) \parallel U\right) \\
&= \sum_{c=1}^{C} \bar{P}_c(\tau) \log \frac{\bar{P}_c(\tau)}{U_c} \\
&= \sum_{c=1}^{C} \left[\mathbb{E}_{x \sim \mathcal{D}_{bc}} P(y = c \mid x; \tau)\right] \log \frac{\mathbb{E}_{x \sim \mathcal{D}_{bc}} P(y = c \mid x; \tau)}{1/C} \\
&= \sum_{c=1}^{C} \left[\mathbb{E}_{x \sim \mathcal{D}_{bc}} P(y = c \mid x; \tau)\right] \left(\log \left[\mathbb{E}_{x \sim \mathcal{D}_{bc}} P(y = c \mid x; \tau)\right] + \log C\right) \\
&= \mathbb{E}_{x \sim \mathcal{D}_{bc}} \sum_{c=1}^{C} P(y = c \mid x; \tau) \log \left[\mathbb{E}_{x' \sim \mathcal{D}_{bc}} P(y = c \mid x'; \tau)\right] + \log C \\
&= \mathbb{E}_{x \sim D_{bc}} \sum_{c=1}^{C} P_c \log P_c + \log C,
\end{aligned}
\tag{10}
$$

where $P_c = P(y = c \mid x; \tau)$ denotes the conditional probability that a sample $x$ belongs to class $c$ under the adjusting of $\tau$.

**Step 2: Gradient Derivation** Taking the derivative of (10) with respect to $\tau$ using the chain rule:

$$
\nabla_\tau \mathcal{J}(\tau) = \mathbb{E}_{x \sim D_{bc}} \sum_{c=1}^{C} \frac{\partial P_c}{\partial \tau} \left(\log P_c + 1\right).
\tag{11}
$$

Given the component $\log P_c + 1 = \theta_c(x) - \tau \cdot \log \pi_c(y) - \log \sum_{k=1}^{C} e^{\theta_k(x) - \tau \cdot \log \pi_k} + 1$ is differentiable in $\tau$, we focus on analyzing the differentiability of $\frac{\partial P_c}{\partial \tau}$. For the softmax function $P_c = \frac{e^{z_c}}{\sum_{k=1}^{C} e^{z_k}}$ with $z_c = \theta_c(x) - \tau \cdot \log \pi_c(y)$, we derive $\frac{\partial P_c}{\partial \tau}$ as follows:

$$
\begin{aligned}
\frac{\partial P_c}{\partial \tau} &= \frac{\partial}{\partial \tau} \left(\frac{e^{z_c}}{\sum_{k=1}^{C} e^{z_k}}\right) \\
&= \frac{\frac{\partial e^{z_c}}{\partial \tau} \cdot \sum_{k=1}^{C} e^{z_k} - e^{z_c} \cdot \frac{\partial}{\partial \tau} \sum_{k=1}^{C} e^{z_k}}{\left(\sum_{k=1}^{C} e^{z_k}\right)^2} \\
&= \frac{e^{z_c} \cdot \frac{\partial z_c}{\partial \tau} \cdot \sum_{k=1}^{C} e^{z_k} - e^{z_c} \cdot \sum_{k=1}^{C} e^{z_k} \cdot \frac{\partial z_k}{\partial \tau}}{\left(\sum_{k=1}^{C} e^{z_k}\right)^2} \\
&= \frac{e^{z_c}}{\sum_{k=1}^{C} e^{z_k}} \cdot \frac{\frac{\partial z_c}{\partial \tau} \cdot \sum_{k=1}^{C} e^{z_k} - \sum_{k=1}^{C} e^{z_k} \cdot \frac{\partial z_k}{\partial \tau}}{\sum_{k=1}^{C} e^{z_k}} \\
&= P_c \cdot \left(\frac{\partial z_c}{\partial \tau} - \sum_{k=1}^{C} \frac{e^{z_k}}{\sum_{k=1}^{C} e^{z_k}} \cdot \frac{\partial z_k}{\partial \tau}\right) \\
&= P_c \cdot \left(\frac{\partial z_c}{\partial \tau} - \sum_{k=1}^{C} P_k \cdot \frac{\partial z_k}{\partial \tau}\right).
\end{aligned}
\tag{12}
$$

Differentiate $z_c = \theta_c(x) - \tau \cdot \log \pi_c(y)$ with respect to $\tau$, we have $\frac{\partial z_c}{\partial \tau} = -\log \pi_c(y)$ and $\frac{\partial z_k}{\partial \tau} = -\log \pi_k$. Substituting $\frac{\partial z_c}{\partial \tau}$ and $\frac{\partial z_k}{\partial \tau}$ into (12), we have:

$$\frac{\partial P_c}{\partial \tau} = P_c \cdot \left( -\log \pi_c(y) - \sum_{k=1}^{C} P_k \cdot (-\log \pi_k) \right)$$

$$= P_c \cdot \left( -\log \pi_c(y) + \sum_{k=1}^{C} P_k \log \pi_k \right). \tag{13}$$

Given the continuous differentiability of $P_c$ and the the constancy of $\pi_c(y)$, the gradient $\frac{\partial P_c}{\partial \tau}$ exists and is differentiable. Substituting (13) back into (11), the gradient $\nabla_\tau J(\tau)$ is thus continuously differentiable, and simplified to:

$$\nabla_\tau J(\tau) = \mathbb{E}_{x \sim D_{\text{bc}}} \sum_{c=1}^{C} P_c \left( -\log \pi_c(y) + \sum_{k=1}^{C} P_k \log \pi_k \right) (\log P_c + 1). \tag{14}$$

Thus, by the differentiability of $\mathcal{J}(\tau)$, gradient descent guarantee convergence to a local minimum. $\square$

*Proof of Lemma 1.* We establish the bounds of $J(\tau)$ as follows. We first expand the outer optimization function according to the definition of KL divergence:

$$J(\tau) = \text{KL}\left( \bar{P}(\tau) \| U \right) = \sum_{c=1}^{C} \bar{P}_c(\tau) \log \left( \frac{\bar{P}_c(\tau)}{U_c} \right)$$

$$= \sum_{c=1}^{C} \bar{P}_c(\tau) \log \left( \frac{\bar{P}_c(\tau)}{\frac{1}{C}} \right)$$

$$= \sum_{c=1}^{C} \bar{P}_c(\tau) \log \left( \bar{P}_c(\tau) \cdot C \right)$$

$$= \sum_{c=1}^{C} \bar{P}_c(\tau) \left[ \log \bar{P}_c(\tau) + \log C \right]$$

$$= \sum_{c=1}^{C} \bar{P}_c(\tau) \log \bar{P}_c(\tau) + \log C \cdot \sum_{c=1}^{C} \bar{P}_c(\tau). \tag{15}$$

Note that $\sum_{c=1}^{C} \bar{P}_c(\tau) = 1$ (since $\bar{P}(\tau)$ is a probability distribution) in the second term. The first term is the negative entropy $\sum_{c=1}^{C} \bar{P}_c(\tau) \log \bar{P}_c(\tau) = -H\left( \bar{P}(\tau) \right)$ of $\bar{P}(\tau)$. Thus, we obtain:

$$J(\tau) = \log C - H\left( \bar{P}(\tau) \right). \tag{16}$$

**Lower Bound** The entropy $H\left( \bar{P}(\tau) \right)$ is maximized when $\bar{P}(\tau)$ is uniform, achieving $H\left( \bar{P}(\tau) \right) = \log C$. Therefore:

$$J(\tau) = \log C - H\left( \bar{P}(\tau) \right) \geq \log C - \log C = 0. \tag{17}$$

Equality holds if and only if $\bar{P}(\tau) = U$. Thus, the lower bound of $J(\tau)$ is 0.

**Upper Bound** The entropy $H\left( \bar{P}(\tau) \right) \geq 0$ for any probability distribution $\bar{P}(\tau)$. Therefore:

$$J(\tau) = \log C - H\left( \bar{P}(\tau) \right) \leq \log C - 0 = \log C. \tag{18}$$

Equality holds when $\bar{P}(\tau)$ is a degenerate distribution (i.e., one class has probability 1 and others 0).

Combining these results, we conclude:

$$0 \leq J(\tau) \leq \log C. \tag{19}$$

$\square$

*Proof of Theorem 2.* We prove the existence of the global minimum point from the perspectives of continuity, lower boundedness and upper asymptotic convergence. Given that we have proven the continuity of $J(\tau)$ in Theorem1, we focus on proving the lower boundedness and upper asymptotic convergence of $J(\tau)$.

**Lower Boundedness** By Lemma 1, $J(\tau) = \log C - H(\bar{P}(\tau)) \geq 0$ for all $\tau \in \mathbb{R}$.

**Upper Asymptotic Convergence** As $\tau \to +\infty$, let $c^* = \arg\min_y \log \pi(y)$. Then, $-\tau \cdot \log \pi(c^*)$ dominates, making:

$$\lim_{\tau \to +\infty} P(y = c \mid x; \tau) = \lim_{\tau \to +\infty} \frac{e^{\theta_c(x) - \tau \cdot \log \pi(c)}}{\sum_{k=1}^{C} e^{\theta_k(x) - \tau \cdot \log \pi(k)}} = \begin{cases} 1, & \text{if } c = c^*, \\ 0, & \text{otherwise.} \end{cases} \quad (20)$$

Consequently, $\bar{P}(\tau) \to \delta_{c^*}$, a Dirac-delta distribution as follows:

$$\begin{aligned} \lim_{\tau \to +\infty} \bar{P}_c(\tau) &= \lim_{\tau \to +\infty} \mathbb{E}_{x \sim \mathcal{D}_{bc}} P(y = c \mid x; \tau) \\ &= \mathbb{E}_{x \sim \mathcal{D}_{bc}} \lim_{\tau \to +\infty} P(y = c \mid x; \tau) \\ &= \begin{cases} 1, & \text{if } c = c^*, \\ 0, & \text{otherwise.} \end{cases} \end{aligned} \quad (21)$$

Substituting $\bar{P}(\tau)$ into $H(\bar{P}(\tau))$, we obtain:

$$\begin{aligned} \lim_{\bar{P}(\tau) \to \delta_{c^*}} H(\bar{P}(\tau)) &= - \lim_{\bar{P}(\tau) \to \delta_{c^*}} \sum_{c=1}^{C} \bar{P}_c(\tau) \log \bar{P}_c(\tau) \\ &= - \sum_{c=1}^{C} \lim_{\bar{P}_c(\tau) \to \delta_{c^*}} \bar{P}_c(\tau) \log \bar{P}_c(\tau). \end{aligned} \quad (22)$$

For the components in 22, when $c = c^*$, $\lim_{\bar{P}_{c^*}(\tau) \to 1} \bar{P}_{c^*}(\tau) \log \bar{P}_{c^*}(\tau) = 0$. When $c \neq c^*$, $\lim_{\tau \to +\infty} \bar{P}_c(\tau) = 0$, $\lim_{\bar{P}_c(\tau) \to 0} \bar{P}_c(\tau) \log \bar{P}_c(\tau) = 0$. Therefore, the addition result of the components is $\lim_{\bar{P}(\tau) \to \delta_{c^*}} H(\bar{P}(\tau)) = 0$. Substituting $H(\bar{P}(\tau)) \to 0$ into $J(\tau)$:

$$\lim_{\tau \to +\infty} J(\tau) = \log C - H(\bar{P}(\tau)) = \log C - 0 = \log C. \quad (23)$$

As $\tau \to -\infty$, define $c^* = \arg\max_y \log \pi(y)$. Similarly, we have:

$$\lim_{\tau \to -\infty} J(\tau) = \log C. \quad (24)$$

$\square$

*Proof of Proposition 1.* The KL divergence $\text{KL}(\bar{P}(\tau) \parallel U)$ is minimized when $\bar{P}(\tau) = U$, as the KL divergence is non-negative and zero at exact matching. This requires solving:

$$\mathbb{E}_{x \sim \mathcal{D}_{bc}} P(x, \theta; \tau) = \mathbb{E}_{x \sim \mathcal{D}_{bc}} \text{softmax}(\theta(x) - \tau \cdot \log \pi(y)) = U, \quad U_c = \frac{1}{C} \; (\forall c \in \{1, \dots, C\}). \quad (25)$$

Leveraging the normalized weighted geometric mean approximation [32], 25 can be approximated as:

$$\begin{aligned} \mathbb{E}_{x \sim \mathcal{D}_{bc}} \text{softmax}(\theta(x) - \tau \cdot \log \pi(y)) &\approx \text{softmax}(\mathbb{E}_{x \sim \mathcal{D}_{bc}}[\theta(x) - \tau \cdot \log \pi(y)]) \\ &= \text{softmax}(\mathbb{E}_{x \sim \mathcal{D}_{bc}}[\theta(x)] - \tau \cdot \log \pi(y)) \qquad \approx U. \quad (26) \end{aligned}$$

Considering the $i$-th component in 26, we have:

$$\text{softmax}(\mathbb{E}_{x \sim \mathcal{D}_{bc}}[\theta_c(x)] - \tau \cdot \log \pi(c)) \approx \frac{1}{C}. \quad (27)$$

Taking the log of both sides, we obtain:

$$\mathbb{E}_{x \sim \mathcal{D}_{bc}}[\theta_c(x)] - \tau \cdot \log \pi(c) = \log K, \quad \forall c, \quad (28)$$

where $K = \frac{\sum_{j=1}^{C} (\exp(\mathbb{E}_{x \sim \mathcal{D}_{bc}}[\theta_j(x)] - \tau \log \pi_j(y)))}{C}$ denotes a constant. Choosing class $c = 1$ as a reference, we subtract the equation for $c = 1$ from that of class $c$:

$$\mathbb{E}_{x \sim \mathcal{D}_{bc}}[\theta_c(x)] - \mathbb{E}_{x \sim \mathcal{D}_{bc}}[\theta_1(x)] = \tau \cdot (\log \pi(c) - \log \pi(1)). \tag{29}$$

Define the residual 29 $r_c = \mathbb{E}_{x \sim \mathcal{D}_{bc}}[\theta_c(x)] - \mathbb{E}_{x \sim \mathcal{D}_{bc}}[\theta_1(x)] - \tau \cdot (\log \pi(c) - \log \pi(1))$, the least-squares objective is:

$$\min_{\tau} \sum_{c=1}^{C} r_c^2 = \min_{\tau} \sum_{c=1}^{C} \left( \mathbb{E}_{x \sim \mathcal{D}_{bc}}[\theta_c(x)] - \mathbb{E}_{x \sim \mathcal{D}_{bc}}[\theta_1(x)] - \tau (\log \pi(c) - \log \pi(1)) \right)^2. \tag{30}$$

To derive the closed-form solution for $\tau$, we define the objective function $f(\tau) = \sum_{c=1}^{C} (\mu_c - \mu_1 - \tau(\log \pi(c) - \log \pi(1)))^2$ with $\mu_c = \mathbb{E}_{x \sim \mathcal{D}_{bc}}[\theta_c(x)], \forall c$. Using the chain rule, we differentiate $f(\tau)$ with respect to $\tau$:

$$\frac{\partial f(\tau)}{\partial \tau} = \sum_{c=1}^{C} 2(\mu_c - \mu_1 - \tau(\log \pi(c) - \log \pi(1))) \cdot \frac{\partial}{\partial \tau} [\mu_c - \mu_1 - \tau(\log \pi(c) - \log \pi(1))]$$

$$= \sum_{c=1}^{C} 2(\mu_c - \mu_1 - \tau(\log \pi(c) - \log \pi(1))) \cdot (-(\log \pi(c) - \log \pi(1)))$$

$$= -2 \sum_{c=1}^{C} (\mu_c - \mu_1 - \tau(\log \pi(c) - \log \pi(1)))(\log \pi(c) - \log \pi(1)). \tag{31}$$

Setting the Derivative to Zero, we obtain:

$$-2 \sum_{c=1}^{C} (\mu_c - \mu_1 - \tau(\log \pi(c) - \log \pi(1)))(\log \pi(c) - \log \pi(1)) = 0$$

$$\sum_{c=1}^{C} (\mu_c - \mu_1)(\log \pi(c) - \log \pi(1)) - \tau \sum_{c=1}^{C} (\log \pi(c) - \log \pi(1))^2 = 0. \tag{32}$$

Rearranging terms to solve for $\tau$:

$$\tau \sum_{c=1}^{C} (\log \pi(c) - \log \pi(1))^2 = \sum_{c=1}^{C} (\mu_c - \mu_1)(\log \pi(c) - \log \pi(1))$$

$$\tau^{\text{LS}} = \frac{\sum_{c=1}^{C} (\mu_c - \mu_1)(\log \pi(c) - \log \pi(1))}{\sum_{c=1}^{C} (\log \pi(c) - \log \pi(1))^2}, \tag{33}$$

where $\tau^{\text{LS}}$ is the final closed-form solution for $\tau$.

$\square$

## A.3 Methods of Label Noise Addition

**Sym.** Symmetric noise (Sym) means that for each sample label, we randomly replace it with one of the other classes with a fixed probability $\eta$. For a $C$-class classification task, given a noise rate $\eta$, the original label $y$ is uniformly changed to other classes except $y$ with the probability $\eta$. Specific noise transition matrix: elements on the diagonal are $1 - \eta$, elements on the off-diagonal are $\eta/(C-1)$.

**Asym.** Asymmetric noise (Asym) simulates the real-world label noise structure. It selects "easily confused" class pairs (such as dog $\leftrightarrow$ wolf) and specifies the transition probability, while the rest remain unchanged. Labels are only replaced between similar classes, and are not randomly mislabeled as other classes. The process of label flipping is related to the quantity of each class. With the noise rate denoted as $\eta$, we establish the following definitions: $T_{ij}(x) = P[\widetilde{Y} = j | Y = i, x] = 1 - \eta$ when $i = j$. Conversely, $T_{ij}(x) = P[\widetilde{Y} = j | Y = i, x] = \frac{n_j}{n - n_i} \eta$. Here, $Y$ and $\widetilde{Y}$ represent the random variables for clean labels and noisy labels, respectively.

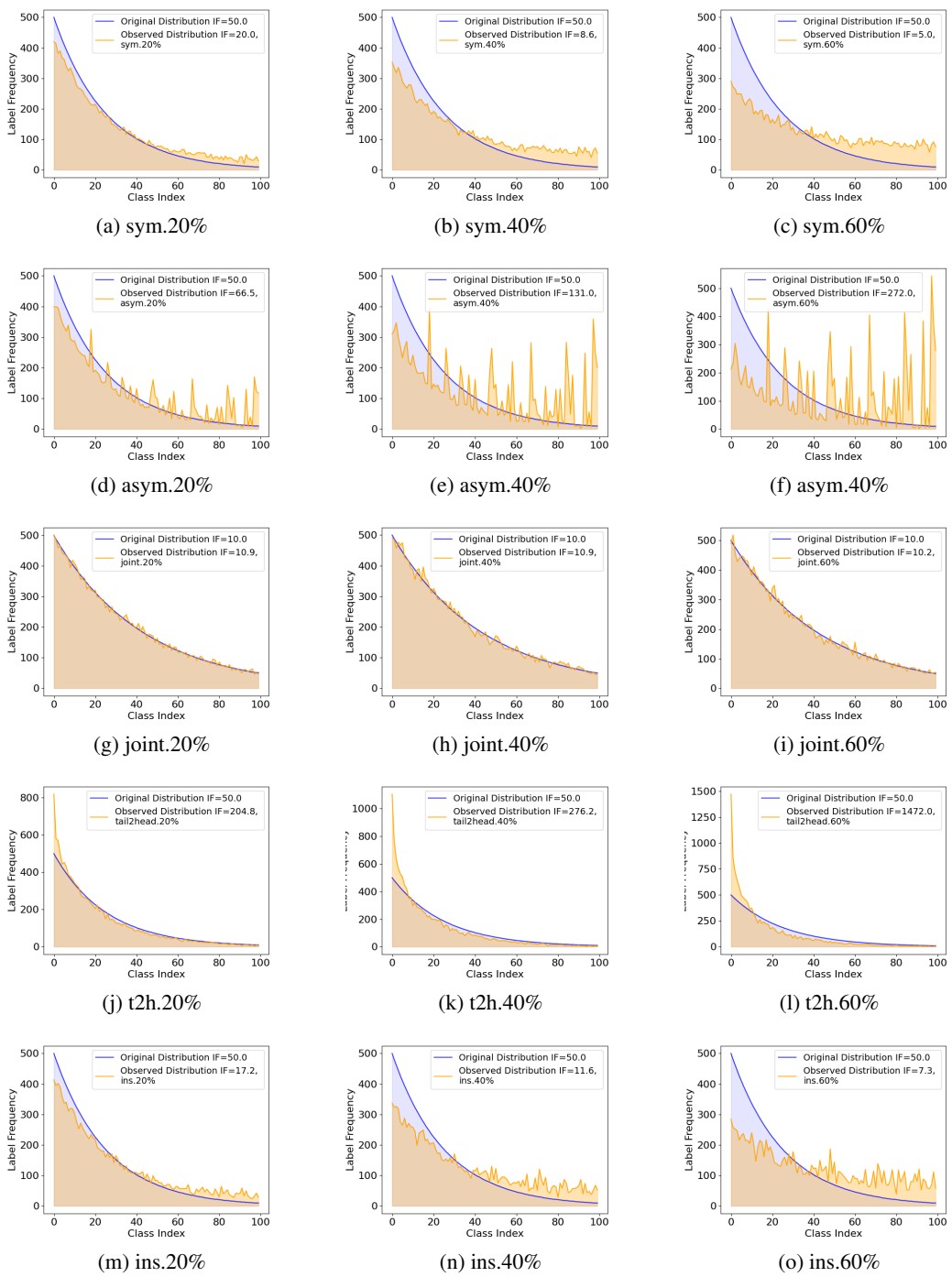

Figure 6: Changes in the imbalance ratio (IR) of observed distributions (orange line) under different noise types with varying noise ratios, in comparison to the fixed IR of the true distritbuion (blue line). Symmetric (sym) and instance-dependent (ins) noise alleviate IR, while asymmetric noise (asym) may reverse the long-tailed pattern. Joint noise preserves IR at low imbalance. Tail-to-head noise (t2h) exacerbates IR compared to the original distribution.

**Joint.** The Joint noise [5] label is generated by the noise transfer matrix, which represents the probability of the clean label flipping to the noise label. Let Y represent the clean label variable, $\tilde{Y}$ represent the noise label, X represent the instance feature, and the transfer matrix $T(X = x)$ is

defined as $T_{ij}(X) = \mathbb{P}(\tilde{Y} = j | Y = i, X = x)$. Specifically, given the noise ratio $\eta \in [0, 1]$, it is defined as follows:

$$T_{ij}(X) = \mathbb{P}\left[\tilde{Y} = j \mid Y = i, X = x\right] = \begin{cases} 1 - \eta & i = j \\ \frac{\eta N_j}{N - N_i} & otherwise \end{cases} \tag{34}$$

Here, N denote the total number of training examples and $N_j$ is the frequency of class $j$. It combines the class prior information in the dataset to set the transition probability, which is more in line with the situation in real-world scenarios where samples are easily mislabeled as frequent classes.

**T2H.** Tail-to-Head noise (T2H) [33] refers to the phenomenon that in the long-tailed data distribution, the tail class samples tend to be mislabeled as the head class samples. The generation of T2H noise mainly includes two steps: separating transferable and non-transferable samples; randomly selecting tail class samples and assigning head class labels to them. We define a transition matrix $T \in \mathbb{R}^{C \times C}$, where each element $T_{t,h} = P(y_i = h | \tilde{y}_i = t)$ represents the probability that an instance with the true class $t$ is mis-labeled as class $h$. In the context of T2H noise, the samples from the tail class $t$ have a relatively high probability of being mis-labeled as the head class $h$, i.e., $T_{t,h} > T_{t,t'}$ (where $t'$ denotes a rarer tail class with fewer samples than $t$). Meanwhile, the probability that a sample from the head class $h$ is mislabeled as the tail class $t$ is very low, $T_{h,t} \approx 0$. Through this transition matrix, the T2H noise is quantitatively defined from a probabilistic perspective.

**IDN.** Instance-dependent noise (IDN) [13] is closely related to the characteristics of each instance and its class label. It is generated by setting a random noise rate for each instance, which follows a truncated Gaussian distribution, and the noise rate of each class is also randomly set. In this noise model, the probability of label flipping varies for each specific instance. Taking the CIFAR-10/100 datasets as examples, when generating instance-dependent noise, for a given clean sample set and a set noise rate $\eta$, a random noise rate is set for each instance one by one according to the truncated gaussian distribution, thereby realizing the generation of instance-dependent noise.

### A.4 Analysis of the Effects of Different Noise Additions Methods on Long-tailed distribution

We systematically investigate the impact of existing noise addition methods on the imbalance ratio (IR) of true distribution when applied to clean long-tailed datasets. The results are presented in the Figure 6. **Sym** noise alleviates the long-tailed problem because the samples of the head category are evenly distributed to other classes, indirectly balancing the data distribution. Baesd on this finding, we choose symmetric noise to simulate *relieve* scenario. **Asym** noise induces a reverse long tail which means that some tail classes surge in count because head-class samples are frequently mislabeled as them. However, the reverse long-tailed scenario is impractical. Therefore, we opt not to use asymmetric noise in our experimental setup. **Joint** noise usually refers to the existence of some correlation between the noisy label and the long-tailed distribution. In some cases like long-tailed distribution with low IR, joint noise maintains the original long-tailed structure, keeping the imbalance ratio unchanged. However, in other cases, it may either exacerbate or alleviate the IR, which is uncontrollable. Thus, we only select joint noise to construct *consistent* scenarios for long-tailed distribution with low IR. **T2H** noise significantly aggravates the long-tailed problem, and the number of tail-class samples is further reduced, causing the model to be biased towards the head classes. We choose t2h noise to construct simulated *aggravate* scenarios. **IDN** noise, like symmetric noise, alleviates IR after adding noise.

### A.5 Results on Simulated Scenarios Based on CIFAR-10

We conduct experiments using the NLL method DPC on CIFAR-10 to evaluate the performance of our method under scenarios with varying imbalance ratios (IR=10, 50, 100), noise types (joint, sym, t2h), and noise rates ($\eta$=40%, 60%). As shown in Table 5, DPC combined with `Unlocker` (DPC+Unlocker) achieves SOTA test accuracies across all the scenarios, outperforming baselines DPC and DPC with direct LA integration. In the consistent scenarios, DPC+Unlocker achieves accuracies of 93.71% and 90.93%, yielding improvements of 0.18% and 5.30% over the original DPC (93.53%, 85.63%) respectively In the relieve scenarios where tail-class clean sample selection is particularly challenging, DPC+Unlocker reaches significant improvements over DPC, ranging from 11.06% to 19.87%, validating its effectiveness in mitigating long tail induced model bias and restoring the original NLL performance. Under aggravate scenarios, DPC+Unlocker maintains stable gains

Table 5: Test accuracy (%) comparison of methods on the CIFAR-10 dataset under varying imbalance ratios (IR), noise types and noise rates $\eta$, involving three scenarios of true distribution shifts. Green numbers indicate improvements over the original NLL method. Boldface represents the best performance in each case.

| dataset | CIFAR-10 | | | | | | | | | |
|---|---|---|---|---|---|---|---|---|---|---|
| types | consistent | | relieve | | | | aggravate | | | |
| IR | 10 | | 50 | | 100 | | 10 | | 50 | |
| $\eta$ | joint 40% | joint 60% | sym 40% | sym 60% | sym 40% | sym 60% | t2h 40% | t2h 60% | t2h 40% | t2h 60% |
| DPC [14] | 93.53 | 85.63 | 78.33 | 57.90 | 60.07 | 45.39 | 93.13 | 72.10 | 74.29 | 70.81 |
| DPC+LA (post-hoc) | 92.74 | 85.31 | 78.93 | 58.05 | 62.40 | 40.08 | 83.82 | 76.21 | 75.93 | 61.12 |
| DPC+LA | 88.31 | 81.53 | 75.60 | 45.77 | 30.94 | 36.50 | 91.27 | 83.11 | 61.75 | 48.40 |
| DPC+LA+Unlocker | **93.71** | **90.93** | **89.39** | **72.86** | **73.15** | **65.26** | **93.76** | **92.05** | **88.67** | **83.35** |
| vs. DPC | ↑0.18 | ↑5.30 | ↑11.06 | ↑14.96 | ↑13.08 | ↑19.87 | ↑0.63 | ↑19.95 | ↑14.38 | ↑12.54 |

over of 1.04% to 19.95% DPC. These results highlight efficacy of `Unlocker` in disentangling the NLL-LTL deadlock and enhancing model robustness against long-tailed noisy label data.

## A.6 Related Work

**Noisy Label Learning (NL).** Noisy label learning focuses on tackling the challenge of inaccurate supervised label in datasets. It mainly evolves along two directions: noisy label detection and correction, and robust noise label learning. The former mainstream typically uses a two-step process: selecting noisy labels via metrics such as loss or divergence, and then correcting them through techniques like semi-supervision learning [10, 11, 12, 13, 14]. By directly selecting and filtering out noisy labels, these methods have demonstrated efficiency in both experimental and real-world scenarios. Robust noisy label learning mitigates noise by adjusting loss functions via regularization or noisy transiton matrix [34, 35, 36, 37, 38] to disregard or reduce noise impact.

**Long-tailed Learning (LT).** Long-tailed learning is aimed to improve the accuracy of tail classes caused by skewed datasets distributions. Re-sampling is a classic method, which directly balances the distribution through reducing samples of head or augmenting samples of tail [39, 40, 41]. Re-weighting enhances the focus of the model on tail classes by adjusting the sample weights in the loss function [42, 23, 43, 44]. Ensembling learning improves model performance by aggregating multiple networks within a multi-expert framework [45, 46, 47, 48, 49]. The two-stage decoupling strategy achieves a rebalancing of decision boundaries through the fine-tuning of classifiers [50, 18, 51]. Logit Adjustment (LA) corrects biased logits by adding an offset term to the model's logit [15, 16, 17, 18, 19]. Extensive empirical studies have substantiated the efficacy of the LA. Moreover, data augmentation is an an effective way to alleviate the scarcity of tail classes by generating tail samples [52, 53, 54]. Besides, recent work such as strategy fusion tailored for multi-objective optimization (MOO) [55] and model parameter space rebalancing [56] also shows promise in balancing model.

