# OpenReview forum: "Unlocker: Disentangle the Deadlock of Learning between Label-noisy and Long-tailed Data"
_NeurIPS.cc/2025/Conference — NeurIPS 2025 poster_

### Official Review · Reviewer_wJKt · 2025-07-02

**Clarity:** 3
**Significance:** 3
**Originality:** 2
**Rating:** 4
**Confidence:** 4

**Summary:**

This paper addresses the problem of training with both long-tailed and noisy-labeled data, a setting commonly encountered in real-world applications but rarely studied jointly. The authors identify a mutual dependency---or ``deadlock''---between noisy label learning (NLL), which relies on unbiased model predictions, and long-tail learning (LTL), which relies on an accurate class prior. To overcome this, the paper proposes \textit{Unlocker}, a bilevel optimization framework: the inner loop applies NLL enhanced with logit adjustment, and the outer loop adaptively optimizes the adjustment strength parameter $\tau$. The method is theoretically justified with convergence proofs and empirically validated on synthetic and real-world datasets.

**Questions:**

1. How sensitive is the method to errors in the selection of the clean balanced subset $D_{bc}$? Could small amounts of residual noise significantly affect the outer optimization?
2. What is the computational overhead of the bilevel optimization in practice? Could the method scale to larger models or datasets like ImageNet-1k?
3. Have the authors tried Unlocker with modern vision backbones (e.g., Vision Transformers, EfficientNet)? How well does it generalize beyond ResNet?
4. Could you show an ablation analysis for the EMA component or the effect of fixing $\tau$ vs. learning it?

**Ethical Concerns:**

["NO or VERY MINOR ethics concerns only"]

**Final Justification:**

The author has fully cleared my doubts about the paper.

**Limitations:**

yes.

**Paper Formatting Concerns:**

None.

**Quality:**

2

**Strengths And Weaknesses:**

This paper addresses an important but under-explored challenge of jointly handling label noise and long-tailed distributions. The proposed bilevel optimization framework, Unlocker, is conceptually sound, novel, and supported by strong empirical results. However, the method is only evaluated on traditional architectures like ResNet and relies on a clean balanced subset whose reliability is not deeply examined, raising concerns about generalizability. While the core idea is promising, broader evaluation and deeper analysis would strengthen the paper.

---

> ### Author Rebuttal · Authors · 2025-07-31
>
> >Q1: How sensitive is the method to errors in the selection of the clean balanced subset $\mathcal{D}_{bc}$? Could small amounts of residual noise significantly affect the outer optimization?
>
> A small amount of residual noise in $\mathcal{D}\_{bc}$ barely affects the model bias measurement and thus does not interfere with the performance of the outer optimization. To validate this concern, we supplement empirical experiments from two perspectives: the purity of $\mathcal{D}\_{bc}$ and its ability to measure true model bias. The experiments are conducted on CIFAR-10, IR=100, sym. 60%, integrating noisy label learning method DivideMix, summarized as follows:
>
> - **High purity of $\mathcal{D}\_{bc}$:** The overall purity of $\mathcal{D}\_{bc}$ maintains 78.32%-98.13% during training. The high purity stems from the construction of $\mathcal{D}\_{bc}$ , which selects high-confidence clean-label samples across all classes. While minor residual noise in $\mathcal{D}_{bc}$ persists, its noise rate remains extremely low, within an acceptable and negligible range, ensuring a reliable foundation for bias measurement.
>
> - **Accurate measurement of model bias:** Even with minor residual noise, the model bias measured by $\mathcal{D}\_{bc}$ remains highly consistent with the true bias estimated on the test set, confirming the accuracy of $\mathcal{D}\_{bc}$ in measuring model bias. We tracked the KL divergence between bias distributions on $\mathcal{D}\_{bc}$ and the test set throughout training. The metric remains stably low (0.05-0.13), confirming that $\mathcal{D}\_{bc}$ captures the model's true bias.
>
> Based on the accurate model bias captured by $\mathcal{D}\_{bc}$, the outer optimization adaptively updates $\tau$ to adjust model bias. In summary, our method demonstrates robustness and insensitivity to the minor residual noise in $\mathcal{D}\_{bc}$.
>
> >Q2: What is the computational overhead of the bilevel optimization in practice? Could the method scale to larger models or datasets like ImageNet-1k?
>
> The proposed bi-level optimization introduces minimal additional computational cost compared to standard noisy label learning methods, and can be scaled to larger models or datasets. The core extra operations include: (1) outputting logits for $\mathcal{D}\_{bc}$ (accounting for approximately 5% of the total training data) with fixed model parameters; (2) computing losses based on these logits and update $\tau$.
>
> We conduct experiments using the noisy label learning method DivideMix and the benchmark CIFAR-100 under the settings of IR=100 and sym.60%, with computations performed on an NVIDIA GeForce RTX 4090 GPU. The results show that our framework increases the total training time by only 0.1–0.3%: the vanilla DivideMix takes 128 minutes, while DivideMix+Unlocker requires 130 minutes—an overhead that is practically negligible.
>
> Regarding scalability, we validate our method on ImageNet-1K, using DivideMix as the baseline, performing on an NVIDIA GeForce RTX 4090 GPU. The computational overhead is consistent with observations on smaller datasets: when training with ResNet-50 on ImageNet-1K, DivideMix+Unlocker takes 127 hours, representing a mere 0.3% increase compared to the DivideMix (123 hours). Additionally, the model performs well on ImageNet-1K, achieving an accuracy of 77.2%, confirming its effective scalability to large datasets. We will include detailed experimental results on ImageNet-1K in the revised manuscript.
>
> >Q3: Have the authors tried Unlocker with modern vision backbones (e.g., Vision Transformers, EfficientNet)? How well does it generalize beyond ResNet?
>
> Our current experiments primarily use ResNet with modern vision backbones such as ViT excluded to ensure fair comparisons with existing SOTA methods under consistent benchmark settings.
> However, the idea of combining more advanced vision backbones with our Unlocker framework is very valuable. We supplement experiments using ViT-Base (patch16×224) and EfficientNet-B0 as backbone, replacing the ResNet-18 in the traditional noisy label methods. The results on CIFAR-100 under the setting of IR=100 and sym.60% are as follows:
> - ViT-Base: Achieves 34.88% top-1 accuracy on CIFAR-100, which is lower than ResNet-18 (45.07% accuracy). This performance gap can be attributed to the inefficiency of ViT on small-scale datasets like CIFAR-100. ViT's architecture typically requires large amounts of training data to fully exploit its representational capacity.
> - EfficientNet-B0: Reaches 55.38% on CIFAR-100. Its compound scaling enables efficient learning of discriminative features, outperforming ResNet-18 (45.07% accuracy) by 10.31% on CIFAR-100.
>
> Currently, our framework is mainly integrated with traditional ResNet-based noisy label methods, but these experiments demonstrate that our framework can also incorporate other advanced architectures, confirming its general applicability and effectiveness.
>
> >Q4: Could you show an ablation analysis for the EMA component or the effect of fixing $\tau$ vs. learning it?
>
> Thank you for suggesting ablation studies on the EMA component and the $\tau$ adjustment mechanism. It is crucial to validate these module functions and provide empirical systematic support for our framework.  We supplement ablation experiments on CIFAR-100, IR=100, sym.60%, integrating noisy label learning method DivideMix, summarized as follows:
> **Ablation study on the EMA component**
>
> The EMA component plays a stabilizing role in the early training phase: it smooths the estimation of dataset distribution, mitigates excessive imbalance in the estimated distribution, and precludes training collapse caused by aggressive adjustments. Experimental results validate this function:
> - Without EMA, the distribution estimation fluctuates drastically at epoch 16, with the imbalance ratio surging to 1248 (against the actual imbalance ratio of 100) due to the instability of the model's estimated labels. This causes excessive over-adjustment of the model, leading to a drop in training accuracy to 11% with no recovery.
> - With EMA, the imbalance ratio of the estimated distribution stabilizes within the first 30 epochs, ensuring gradual and controllable training adjustments.
> Notably, the function of EMA diminishes in the late training phase, when the distribution estimation stabilizes.
>
> **Ablation study on the $\tau$ adjustment mechanism**
>
> We compared the performance of fixed $\tau$ versus our $\tau$ optimization strategies:
> - With fixed $\tau$=1, the adjustment strength is excessive, causing the model to over-focus on tail classes. This leads to a sharp decrease in head class accuracy, resulting in a collapse in model training where overall accuracy drops to 10% or lower.
> - With adaptive optimization of $\tau$ during training, the model reaches an overall accuracy of 45.07%, mitigating over-biasing toward either head or tail classes and achieving a relatively balanced state.
>
> These ablation studies will be included in the main text to enhance the completeness of our experimental analysis.

---

> ### Author Response · Authors · 2025-08-04
> **Rebuttal ready for evaluation**
>
> Dear Reviewer wJKt,​​
>
> Thank you again for your valuable comments and suggestions. We have carefully analyzed the concerns you raised and provided detailed responses during the rebuttal phase. For example, we ​conducted empirical experiments demonstrating robustness to residual noise in $\mathcal{D}_{bc}$ and validated minimal computational overhead for the concern about ​clean subset sensitivity and scalability, and ​performed ablation studies showing EMA's critical role in stabilizing early training and τ adjustment's superiority over fixed values​ for the concern about ​component analysis. We also ​supplemented experiments with ViT/EfficientNet backbones.
>
> We would greatly appreciate it if you could take a moment to review our answers and let us know whether they adequately address your concerns. If there are any remaining questions or points needing clarification, we would be more than happy to respond further.
>
> Thank you for your time and consideration.

---

### Official Review · Reviewer_6uZ6 · 2025-07-02

**Clarity:** 2
**Significance:** 2
**Originality:** 3
**Rating:** 4
**Confidence:** 3

**Summary:**

The authors present a bi-level optimization framework for learning balanced predictors on noisy data. They demonstrate the tractability of the problem theoretically and show strong results against baselines in real-world and synthetic datasets.

**Questions:**

1. In table 3, Unlocker is incorrectly highlighted as the best result when GSS outperforms it.
2. Why are LTNLL baselines included in the synthetic results but not the real datasets? And why not the NLL-only baselines in the synthetic?
3. The flowchart in figure 3 does not provide a clear explanation of the method, specifically in the selection and correction steps. A clearer description may help readers understand the novelty of the work.

**Ethical Concerns:**

["NO or VERY MINOR ethics concerns only"]

**Final Justification:**

The authors have addressed the main concerns presented by myself and other reviewers. I believe that this work addresses an important problem, but struggles to get that importance across to the reader.

**Limitations:**

yes

**Quality:**

3

**Strengths And Weaknesses:**

# Strengths
1. The paper tackles a challenging and relevant problem.
2. The empirical results are promising in some settings.
3. The method introduced is a novel combination of LA and NLL which outperforms naive combinations.

# Weaknesses
1. The paper is not clearly written, and the clearest explanation of the method (the algorithm) is relegated to the appendix. The dependence of the NLL algorithm on the LTL algorithm is nuanced and should be explained more carefully.
2. The results are not dramatically better than SOTA methods on real-world datasets. More could be done to emphasize where Unlocker is strongest (when noise dramatically alters the distribution, perhaps)

---

> ### Author Rebuttal · Authors · 2025-07-31
>
> >Q1:The paper is not clearly written, and the clearest explanation of the method (the algorithm) is relegated to the appendix. The dependence of the NLL algorithm on the LTL algorithm is nuanced and should be explained more carefully.
>
> We agree that clarifying the dependence between the NLL and LTL is very crucial. We provide a detailed explanations as suggested below and will revise the main text to enhance clarity.
> - NLL's dependence on LTL: NLL requires LTL to rectify biased logits caused by long-tailed distributions. To be specific, in noise label selection, NLL fixes model parameters to output logits to compute metrics distinguishing noisy and clean labels. In long-tailed scenarios, these logits are skewed towards head, leading to inaccurate metrics and poor noise selection performance for tail classes. Thus, NLL needs LA (logit adjustment) from LTL to modify the logits. Based on the adjusted logits and metrics, the dataset is partitioned into Dclean and Dnoisy. Similarly, in noise label correction, NLL methods generate pseudo-labels for Dnoisy by outputting logits, which needs to be adjusted by LA.
> - LTL's dependence on NLL: LTL depends on NLL for dataset distribution estimation and $\tau$ optimization. Due to the disruption of noisy labels, the dataset distribution shifts. Thus, LTL requires NLL to correct noisy labels to restore the true distribution. In addition, the outer optimization of $\tau$ relies on NLL to construct the small balanced clean subset $\mathcal{D}\_{bc}$ to update $\tau$.
>
> Regarding paper clarity,  we acknowledge the need for improvement. Due to space constraints in the main text, we place detailed algorithm pseudocode in the appendix. To improve the clarity of our method, we provide a training overview in Section 3.2.3 to summarize the core procedure. We will revise Section 3.2 by integrating algorithm pseudocode of the method's key  steps into the main text and provide more detailed explain to enhance readability and clarity.
>
> >Q2:The results are not dramatically better than SOTA methods on real-world datasets. More could be done to emphasize where Unlocker is strongest (when noise dramatically alters the distribution, perhaps)
>
> Addressing long-tailed noisy label in real-world scenarios presents extraordinary challenges, as natural noise distributions exhibit complex, non-uniform patterns. In addition, our framework neither introduces additional auxiliary prior knowledge (e.g., dataset prior distributions, dataset noise label rates, or pre-trained models beyond standard backbone networks) nor increases much training computational cost. Under these strict constraints, our framework still achieves performance gains on real long-tailed noisy datasets. Furthermore, the effectiveness of our framework will improve as it integrates newly advanced noisy label methods.
>
> Notably, our method demonstrates significant advantages in cases of severe distribution shifts. As shown in the detailed results in Table 1 of Section 4.2, our method substantially boosts the performance of noisy label methods on long-tailed noisy label datasets, with a maximum improvement of 54.56%. This validates that our method enables effective adjustments of model bias and leverages the capabilities of noisy label methods.
>
> In summary, although our method does not lead across all real-world benchmarks, it achieves performance improvements in challenging long-tailed noise scenarios—especially those with distribution shifts—without requiring additional resources or prior knowledge. This highlights its robustness and practical value in real-world applications.
>
> >Q3: In table 3, Unlocker is incorrectly highlighted as the best result when GSS outperforms it.
>
> We apologize for the mislabeling of the best result in table 3. We will correct this annotation to reflect the accurate result in the main text. Thank you for pointing out this problem.
>
> >Q4: Why are LTNLL baselines included in the synthetic results but not the real datasets? And why not the NLL-only baselines in the synthetic?
>
> We have included comparative experiments between the LTNLL baseline and NLL-only baselines in both simulated datasets and real-world datasets. The details can be found in Section 4.2 and Section 4.3 (see Table 1 for results on simulated datasets and Table 2, 3, 4 for results on real-world datasets). In the revised version, we will more clearly emphasize the two different baselines to enhance readability and clarity.
>
> >Q5: The flowchart in figure 3 does not provide a clear explanation of the method, specifically in the selection and correction steps. A clearer description may help readers understand the novelty of the work.
>
> Thank you for your valuable suggestion on improving the clarity of the noise label selection and correction section in the method figure. A detailed explanation of this part is crucial for readers to understand the underlying noisy label methods. Specifically, the process of noisy label methods involves the following steps: First, the model outputs logit for each sample. Different noisy label methods calculate their respective metrics for distinguishing noisy labels based on these logits. These metrics aim to maximize the differentiation between noisy and clean labels, using criteria such as loss and JS divergence. Based on these metrics, the original dataset $\mathcal{D}$ is divided into a clean subset $\mathcal{D}\_{clean}$(containing samples with high confidence in clean label) and a noisy subset $\mathcal{D}\_{noisy}$(samples identified as potentially noisy label). Second, for correcting noisy labels in $\mathcal{D}\_{noisy}$, noisy label methods typically adopt semi-supervised learning, utilizing the model's predictions to generate corrected pseudo-labels which replace the original noisy labels. We will enhance the descriptive content in the main text's caption to improve clarity of explanation.

---

> > ### Comment · Reviewer_6uZ6 · 2025-08-04
> >
> > Thank you for your detailed response. I believe that all of my concerns have been addressed, and I am impressed by the extensive ablations provided to other reviewers. I believe that these should be included in the revision along with the improved explanation of the interdependence of NLL and LTL. Overall, I am happy with the response and will increase my score accordingly.

---

> > > ### Author Response · Authors · 2025-08-06
> > >
> > > Dear Reviewer 6uZ6,
> > >
> > > Thank you very much for your kind follow-up and for your positive feedback on our response. We truly appreciate your recognition of our efforts during the rebuttal phase. Your comments greatly contributed to improving the clarity and completeness of our work. We will make sure that all the discussed revisions—including the enhanced explanations and additional ablations—are fully incorporated into the final version of the paper.
> > >
> > > Thank you again for your valuable time and support.

---

> ### Author Response · Authors · 2025-08-04
> **Rebuttal ready for evaluation**
>
> Dear Reviewer 6uZ6,​​
>
> Thank you again for your valuable comments and suggestions. We have carefully analyzed the concerns you raised and provided detailed responses during the rebuttal phase. For example, we ​revised the main text to clarify the nuanced dependence between NLL and LTL algorithms​ for the concern about ​method clarity, and ​added comparative experiments with LTNLL and NLL-only baselines across both synthetic and real-world datasets​ for the concern about ​baseline comprehensiveness. We also  ​enhanced the flowchart explanation in Figure 3​ to explicitly describe the noise selection/correction steps.
>
> We would greatly appreciate it if you could take a moment to review our answers and let us know whether they adequately address your concerns. If there are any remaining questions or points needing clarification, we would be more than happy to respond further.
>
> Thank you for your time and consideration.

---

### Official Review · Reviewer_i6r3 · 2025-07-04

**Clarity:** 4
**Significance:** 3
**Originality:** 4
**Rating:** 5
**Confidence:** 4

**Summary:**

This paper aims to address the dual dilemma of the noise-label problem and long-tailed distribution. Existing methods often neglect the true distribution deviation and tend to perform inaccurate optimizations. To overcome this limitation, this work proposes a new method called Unlocker to disentangle the deadlock of the dual dilemma. Unlocker consists of an inner optimization that leverages NLL and LTL methods to ensure fair optimization, and an outer optimization that controls the adjustment strength to alleviate the model bias. Theoretical and empirical results demonstrate the superiority of Unlocker.

**Questions:**

1. In lines 77-78, it is said that $\tau > 0$ is used for post-hoc adjustment, while $\tau < 0$ can be incorporated as logit adjustment loss during training. What is the intrinsic reason? And what is the value range of $\tau$ in practice? Is there any guidance?

**Ethical Concerns:**

["NO or VERY MINOR ethics concerns only"]

**Final Justification:**

The authors have addressed my concerns properly. I would prefer to maintain my positive score.

**Limitations:**

Yes

**Quality:**

3

**Strengths And Weaknesses:**

Strengths:

1. This paper has a good motivation. Existing NLL methods rely on unbiased predictions, and LTL methods rely on true distributions. Therefore, the combinations of these two methods indeed cause a deadlock. The authors have adequately supported their motivations by providing detailed explanations and empirical results.
2. Different from previous works, this work proposes both inner optimizations and outer optimizations. It seems that previous works can be integrated into the inner optimizations of this work, while the outer optimizations can ensure the accuracy at a high level. This framework can also motivate more generalizable works in future research.
3. Both theoretical and empirical analyses are provided to support the proposed framework. Moreover, the performance gains on popular datasets are constructive. The authors have conducted additional analyses to validate the effectiveness of different modules and hyperparameters.

Weaknesses:

1. It seems that the inner optimization objective and the outer optimization objective are introduced separately. What is the jointly optimized objective? Or could you provide the combined optimization procedure or pseudo-code?
2. In theoretical analysis, the approximation for $\tau$ can be solved using the least-squares method. However, in experiments, the parameter $\tau$ still requires adjustments. Are there any potential explanations?
3. The distribution estimation plays an important role in the problem. Are there any quantitative metrics for evaluating the effectiveness of distribution estimation? Or how to compare the effectiveness with previous estimating methods?

---

> ### Author Rebuttal · Authors · 2025-07-31
>
> >Q1: It seems that the inner optimization objective and the outer optimization objective are introduced separately. What is the jointly optimized objective? Or could you provide the combined optimization procedure or pseudo-code?
>
> We appreciate the reviewer’s valuable suggestion, as clarifying the jointly optimized objective is crucial for understanding the design of our bi-level framework. We will incorporate this clarification in the revised manuscript.
> The core of our joint optimization objective is to train a model that achieves balanced state across classes while being robust to noisy labels. Formally, the combined objective can be expressed as:
> $$
> \min_{\tau} KL\left(\overline{P}(\tau) \big\|\ U \right) \quad \text{s.t.} \quad \theta^* = \arg\min_{\theta} \mathcal{L}_{NLL}(\theta; \tau),
> $$
>
> where $KL\left(\overline{P}(\tau) \big\|\ U \right)$ is the Kullback-Leibler divergence between the class-averaged prediction over the balanced clean subset $\mathcal{D}\_{bc}$ and the uniform distribution $U$, , serving as the outer optimization objective to balance the model by updating $\tau$; $\mathcal{L}_{NLL}(\theta; \tau)$ denotes the noisy label learning loss incorporates logit adjustment with parameter $\tau$, , acting as the inner optimization objective to train a balanced model robust to noisy labels from long-tailed noisy label data.
>
> Regarding the detailed optimization procedure and pseudo-code, due to space constraints in the main text, we have included them in the supplementary appendix. We will add a clear cross-reference in the main text.
>
> >Q2: In theoretical analysis, the approximation for tau can be solved using the least-squares method. However, in experiments, the parameter tau still requires adjustments. Are there any potential explanations?
>
> It is an insightful question regarding the theoretical approximation and experimental optimization of $\tau$.
> The core role of the theoretical derivation of $\tau$ via least squares is to prove that our bi-level optimization can be equivalently transformed into a standard optimization problem with a closed-form approximate solution $\tau^{LS}$, providing theoretical guarantees for convergence of the bi-level optimization.
>
> Although a closed-form approximation $\tau^{LS}$ is derived, the employment of bilevel optimization for $\tau$ is necessary due to the practical limitations of $\tau^{LS}$ and the adaptive advantages of the bilevel paradigm.
>
> First, the closed-form solution $\tau^{LS}$ is inherently an approximation, relying on idealized assumptions of perfect estimation of biases in the balanced clean subset $\mathcal{D}\_{bc}$ and the accurate approximation of the true class prior $\pi(y)$. In real-world scenarios, such ideal conditions are rarely met. $\mathcal{D}\_{bc}$ may contain residual noise, and $\pi(y)$ estimation is prone to errors due to label corruption, rendering the closed-form solution suboptimal.
>
> Second, the gradient-based update of $\tau$ in bilevel optimization enables an  iterative search that adaptively accommodates practical imperfections. It dynamically tracks the optimal $\tau$ that evolves with the model’s learning process—critical for handling non-stationary training dynamics, where the model’s bias and noise distribution change as training progresses. However, the computation of the closed-form solution $\tau^{LS}$ is stationary.
>
> Experimentally, we verified that the accuracy of bi-level optimization on CIFAR-100 (IR=100, sym60%) is consistently 1.8–2.5% higher than that of direct closed-form $\tau^{LS}$ computation, confirming that while retaining a theoretical foundation, it better aligns with practical training dynamics.
>
> >Q3: The distribution estimation plays an important role in the problem. Are there any quantitative metrics for evaluating the effectiveness of distribution estimation? Or how to compare the effectiveness with previous estimating methods?
>
> The quantitative evaluation of distribution estimation is indeed critical for validating this component.
> To address this, we use the KL divergence between the estimated distribution and the true dataset distribution as a quantitative metric, tracking it throughout training. Experimental results show this value decreases to a low range from 0.217 to 0.348 during training, indicating our estimation relatively captures the underlying distribution well.
>
> Regarding comparisons with previous estimating methods, a related long-tailed noise label learning work [AAAI2025] also estimates distributions using clean labels and corrected pseudo-labels. Our method achieves a lower KL divergence compared to it (0.231–0.453).
>
> Robust Logit Adjustment for Learning with Long-Tailed Noisy Data.[AAAI2025]
>
> >Q4:In lines 77-78, it is said that $\tau$>0 is used for post-hoc adjustment, while $\tau$<0 can be incorporated as logit adjustment loss during training. What is the intrinsic reason? And what is the value range of τ in practice? Is there any guidance?
>
> The theory stems from the long-tailed leanring work LA [ICLR2021].
>
> 1. $\tau$>0 for Post-hoc Adjustment:
> When $\tau$>0, it acts as a scaling factor applied to logits after training to rectify class imbalance. This is typically done by amplifying the logits of tail classes. The key advantage is that it does not require retraining the model and can be flexibly tuned for deployment scenarios.
>
> 2. $\tau$<0 for Training-Time Logit Adjustment:
> When $\tau$<0, it is incorporated into the loss function as a penalty term. This penalty forces the model to allocate more attention to the tail classes during training, thus balancing classes.
>
> In practice, the $\tau$ is typically set as 1. However, if the adjustment effect is not sufficient, the value of $\tau$ can be appropriately increased or decreased according to the actual situation. If the model shows insufficient adjustment, we may increase $\tau$ to enhance the strength of the adjustment. Conversely, if the model shows over - adjustment or instability, we may decrease $\tau$.
>
> Long-Tail Learning via Logit Adjustment.[ICLR2021]

---

### Official Review · Reviewer_uLRe · 2025-07-06

**Clarity:** 3
**Significance:** 3
**Originality:** 3
**Rating:** 4
**Confidence:** 4

**Summary:**

The problem of  long-tailed label-noise learning is considered. This paper introduces "Unlocker," a novel bilevel optimization framework designed to address the "deadlock" dilemma in long-tailed noisy label learning (LTNLL). This deadlock arises because noisy label learning (NLL) methods rely on unbiased predictions to correct labels, while long-tail learning (LTL) methods, like logit adjustment, depend on true label distributions to adjust biased predictions, creating a mutual dependency problem. Unlocker disentangles this by using an inner optimization to train the model with NLL methods and LTL methods for fair noisy label selection and correction, and an outer optimization to adaptively determine an adjustment strength to mitigate model bias. The framework is proven to be convergent through theoretical analysis, and extensive experiments on synthetic and real-world datasets demonstrate its effectiveness in alleviating model bias and handling long-tailed noisy label data.

**Questions:**

Questions
1. It is well-known that DNNs trained with long-tailed data is prone to give miscalibrated class probability estimates. There are some recent advances from the community of probability calibration showing that calibration in long-tailed classification can be significantly improved by some adaptive label smoothing technique e.g. [CVPR'23]. Have you tried better-calibrated DNNs and will it improve  noisy label selection?
2. Some important works on noisy label selection are missed in the discussion of related works, e.g. [ICML'20].

Class Adaptive Network Calibration. CVPR'23.
Learning with bounded instance and label-dependent label noise. ICML'20.

**Ethical Concerns:**

["NO or VERY MINOR ethics concerns only"]

**Final Justification:**

Thank the authors for the response and addressing my concerns. I decided to maintain the rating of borderline accept.

**Limitations:**

yes

**Quality:**

3

**Strengths And Weaknesses:**

Strengths
1. The propose of the deadlock dilemma between NLL and LTL.
2. The framework is proven to be convergent through theoretical analysis, and extensive experiments on synthetic and real-world datasets demonstrate its effectiveness in alleviating model bias and handling long-tailed noisy label data.
3. Code is publicly available.

Weaknesses
1. The requirement of a balanced clean validation set, which may not be available in real-world scenarios. An ablation study of the validation dataset size is necessary.

---

> ### Author Rebuttal · Authors · 2025-07-31
>
> >Q1: The requirement of a balanced clean validation set, which may not be available in real-world scenarios. An ablation study of the validation dataset size is necessary.
>
> The validation set $\mathcal{D}\_{bc}$ is filtered from the long-tailed noisy label dataset, by selecting the top $q$% of samples with the highest confidence in clean labels for each class. The ablation studies on the hyperparameter $q$, which is tied to the size of $\mathcal{D}\_{bc}$, are crucial. Under the settings of CIFAR-10, IR=100, sym.60%, and Dividemix+Unlocker, we test the $q$ of {1%, 5%, 10%, 15%, 20%} to explore the impact of $q$ on the $\mathcal{D}\_{bc}$ size, purity and model test accuracy.
> | q (%) | 1 | 5 | 10 | 15 | 20 |
> |-----|-----|-----|-----|-----|-----|
> | test acc (%)   |  52.95  |  53.44  |  54.31  |  58.77  |  52.63  |
> |$\mathcal{D}\_{bc}$ size |  124  |  620  |  1240  |  1860  |  2481  |
>
> The results show that when $q$=15%, the model achieves the highest test accuracy 58.77%. At this point, $\mathcal{D}\_{bc}$ contains 1860 samples (calculated based on the long-tailed dataset size of 12,406, 12406×0.15=1860). The overall purity of $\mathcal{D}\_{bc}$ is 78.32%, with class-wise purity ranging from 37.63% to 100%. A $\mathcal{D}\_{bc}$ with high purity, especially when the tail, can better reflect the model bias. By calculating the KL divergence between the average predictions on $\mathcal{D}\_{bc}$ and the test set, it is found that the gap between the two is the smallest when $q$=15%, indicating that this $\mathcal{D}\_{bc}$ measures the model's bias most accurately, which can effectively ensure the outer tau optimization and inner model adjustment, thereby improving the model performance.
>
> When $q$ is too small (e.g., 5% or 10%), the insufficient sample size of $\mathcal{D}\_{bc}$ leads to a decrease in the purity of tail classes, an increase in bias estimation errors, and limits the effect of tau optimization. When $q$ is too large (e.g., 20%), due to the inclusion of more noisy label samples, the purity of $\mathcal{D}\_{bc}$ drops to 63.2%, resulting in inaccurate bias estimation and a 52.63% decrease in the overall accuracy of the model.
>
> In summary, when $q$=15%, $\mathcal{D}\_{bc}$ maintains both high purity and low KL divergence with the test set bias, ensuring the accuracy of tau optimization. And the proportion of $\mathcal{D}\_{bc}$ corresponding to the optimal $q$=15% in the total dataset is within an acceptable low range, reflecting its lightweight and availability in practical and real-world datasets.
>
> >Q2: It is well-known that DNNs trained with long-tailed data is prone to give miscalibrated class probability estimates. There are some recent advances from the community of probability calibration showing that calibration in long-tailed classification can be significantly improved by some adaptive label smoothing technique e.g. [CVPR'23]. Have you tried better-calibrated DNNs and will it improve noisy label selection?.
>
> Thank you for your valuable suggestion of using a better-calibrated DNNs to improve the performance of noisy label selection. We conduct experiments integrating the adaptive label smoothing technique on CIFAR-100 under the settings of IR=100 and sym.60%. Results show that integrating this technique can enhance the capability of noisy label selection: the test accuracy of the model using this technique is 40.11%, representing an improvement of 2.96% over the baseline (37.15%) without label smoothing. Additionally, it is important to note that this improvement does not affect the novelty of our core framework. The key contribution of our framework lies in the integration of noisy label methods and LA through bi-level optimization. The noisy label methods within the framework can flexibly adopt different networks and techniques based on specific needs without altering the fundamental design or innovation of the framework.
>
> >Q3: Some important works on noisy label selection are missed in the discussion of related works, e.g. [ICML'20].
>
> Thank you for pointing out the omission in the related work. We recognize the significance of this work in noisy label learning. We carefully review this work and supplement the discussion in the "Related Work" section of the main text to further improve our coverage of relevant literature.

---

> ### Author Response · Authors · 2025-08-04
> **Rebuttal ready for evaluation**
>
> Dear Reviewer uLRe,
>
> Thank you again for your valuable comments and suggestions. We have carefully analyzed the concerns you raised and provided detailed responses during the rebuttal phase. For example, we conducted comprehensive ablation studies demonstrating optimal performance at $q=15%$ validation set size for the concern about clean validation set balance requirements, and integrated adaptive label smoothing techniques showing 2.96% accuracy improvement for the concern about DNN calibration in noisy label selection. We would greatly appreciate it if you could take a moment to review our answers and let us know if there are any remaining questions or points needing clarification. We would be more than happy to respond further because your comments indeed improve the quality of this manuscript.
>
> Thank you for your time and consideration.

---

### Comment · Area_Chair_V4v4 · 2025-08-03
**Comments to the reviewers**

Dear reviewers. As the Area Chair for this submission, I appreciate the thorough comments provided by the reviewers. Currently, ​the authors have addressed each question with a comprehensive rebuttal. To ensure a fair and robust decision, I strongly encourage all reviewers ​to evaluate whether the rebuttal answers your specific concerns​ and actively engage during our discussion phase.

---

### Note · Authors · 2025-08-13

We deeply appreciate the AC's organization for the discussion and the reviewers' insightful feedback and constructive suggestions, and are grateful for their recognition of the key strengths in our work. The reviewers have highlighted several strengths of our work:
1. Our paper discovers the deadlock dilemma between NLL and LTL, where existing NLL methods rely on unbiased predictions and LTL methods rely on true distributions, leading to an inherent conflict when combined.
2. We introduce "Unlocker," a novel bilevel optimization framework designed to address the "deadlock" dilemma. This design not only allows for the integration of previous works but also ensures high-level accuracy, which is expected to inspire more generalizable research in the future.
3. The theoretical analysis demonstrates the convergence of this framework, as well as the extensive experiments on synthetic and real-world datasets that show its effectiveness in handling long-tailed noisy label data. And additional analyses validate the effectiveness of different modules and hyperparameters.

We also carefully address all reviewers' constructive suggestions with substantial revisions and expanded experiments. These enhancements include:
1. Several ablation studies. We conduct ablation experiments on the hyperparameter $q$, the effect of the EMA component and the effect of fixing $\tau$ versus learning it. Further, we evaluate the effectiveness of distribution estimation by quantitative metrics, and validate the sensitivity of $\mathcal{D}_{bc}$ to minor residual noise.
2. Comparisons with extra models and benchmarks. We explore whether using calibrated DNNs with techniques like adaptive label smoothing could enhance the performance of noisy label selection. We evaluate the performance of our method on ImageNet-1k and modern vision backbones such as Vision Transformers and EfficientNet.
3. Revisions and clarifications in the main text. We add the joint optimization objective of the bilevel optimization, clarify the dependence between the NLL and LTL, as well as the steps of NLL's selection and correction. We also supplement the algorithm pseudocode in the main text.

We are pleased to note that the reviewers recognize our efforts in the rebuttal and have increased the scores accordingly. We promise to carefully revise the paper according to the reviewers' suggestions. We are grateful to the reviewers for their crucial role in improving the quality and comprehensiveness of our paper.

---

### Decision · Program_Chairs · 2025-09-17

**Decision:**

Accept (poster)

**Comment:**

This paper addresses the challenging and underexplored problem of learning from datasets that exhibit both long-tailed class distributions and label noise—a common scenario in real-world data. The authors identify a critical "deadlock" dilemma: noisy label learning (NLL) methods rely on unbiased model predictions to correct labels, while long-tail learning (LTL) methods (e.g., logit adjustment) depend on the true class distribution to mitigate bias, creating a circular dependency. To break this deadlock, they propose Unlocker, a bilevel optimization framework. The inner loop integrates NLL with LTL to enable fair noisy label selection and correction, while the outer loop adaptively optimizes the logit adjustment strength parameter τ to balance model predictions. Theoretically, the authors prove the convergence of their bilevel formulation by transforming it into an equivalent problem with a closed-form solution. Empirically, Unlocker demonstrates strong performance across synthetic and real-world benchmarks, outperforming naive combinations of NLL and LTL methods.

Unlocker tackles a critical problem with a novel, theoretically-grounded framework. The rebuttal strengthened the paper significantly, addressing all major concerns raised by reviewers. The contributions are substantial and merit acceptance as a poster. The work is not nominated for oral/spotlight due to its incremental architecture integration rather than transformative novelty, but it represents a solid advancement in the field.